# FAITHEVAL: CAN YOUR LANGUAGE MODEL STAY FAITHFUL TO CONTEXT, EVEN IF "THE MOON IS MADE OF MARSHMALLOWS"

**Yifei Ming**[1]*, **Senthil Purushwalkam**[1], **Shrey Pandit**[2], **Zixuan Ke**[1], **Xuan-Phi Nguyen**[1]
**Caiming Xiong**[1], **Shafiq Joty**[1]
[1]Salesforce AI Research  [2]University of Texas at Austin

## ABSTRACT

Ensuring faithfulness to context in large language models (LLMs) and retrieval-augmented generation (RAG) systems is crucial for reliable deployment in real-world applications, as incorrect or unsupported information can erode user trust. Despite advancements on standard benchmarks, faithfulness hallucination—where models generate responses misaligned with the provided context—remains a significant challenge. In this work, we introduce FaithEval, a novel and comprehensive benchmark tailored to evaluate the faithfulness of LLMs in contextual scenarios across three diverse tasks: unanswerable, inconsistent, and counterfactual contexts. These tasks simulate real-world challenges where retrieval mechanisms may surface incomplete, contradictory, or fabricated information. FaithEval comprises 4.9K high-quality problems in total, validated through a rigorous four-stage context construction and validation framework, employing both LLM-based auto-evaluation and human validation. Our extensive study across a wide range of open-source and proprietary models reveals that even state-of-the-art models often struggle to remain faithful to the given context, and that larger models do not necessarily exhibit improved faithfulness. Code is available at: https://github.com/SalesforceAIResearch/FaithEval.

## 1 INTRODUCTION

The rapid development of large language models (LLMs) has significantly advanced natural language understanding and generation tasks, enabling systems to produce fluent and coherent responses across a variety of applications (Bubeck et al., 2023; Zhao et al., 2024b; Wu et al., 2024b). The capabilities of these models have been further enhanced by integrating external information from the Internet or knowledge sources using a popular approach of Retrieval-Augmented Generation (RAG) (Lewis et al., 2020; Zhao et al., 2024a). In this paradigm, generated outputs are enhanced by retrieving and encoding relevant information as context to the model. While RAG facilitates the integration of additional knowledge, hallucination—where models generate unsupported or ungrounded content—remains a critical challenge (Nguyen et al., 2024).

Hallucination in LLMs can be generally categorized into two types: *factual hallucination*, where generated content deviates from established world knowledge, and *faithfulness hallucination*, where the generated response is inconsistent with the provided context (Huang et al., 2023a). While factuality has received extensive attention, with numerous benchmarks designed to evaluate correctness against common sense or world knowledge (Lee et al., 2022; Min et al., 2023; Chern et al., 2023; Wei et al., 2024), a fine-grained and holistic evaluation of faithfulness on noisy contexts remains underexplored, particularly when the context contradicts commonly accepted facts. Maintaining faithfulness to the context is especially important for various personalized applications and can be critical in high-stakes domains such as healthcare, finance and law, where inaccurate or ungrounded responses can erode user trust and lead to severe consequences (Bommarito & Katz, 2022; Pal et al., 2023).

One of the key challenges in addressing faithfulness hallucination in RAG stems from the retrieval process, where the wealth of documents on the Internet varies in credibility. This complexity is

---

*Correspondence: yifei.ming@salesforce.com

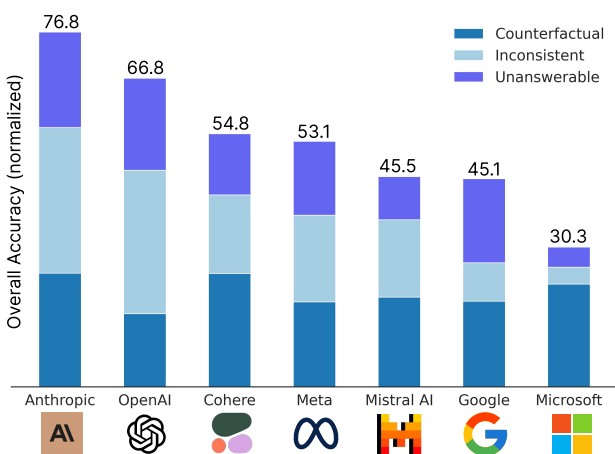

Figure 1: Performance summary on FaithEval Benchmark. Each bar shows the combined accuracy (normalized) for the best model from each organization across three tasks: Counterfactual, Inconsistent, and Unanswerable (Sec 3.1). Different colors in each bar represent the accuracy for each task.

further compounded when long retrieved content includes multiple relevant paragraphs that omit key details, present conflicting evidence, or propagate counterfactual claims. Existing hallucination evaluation benchmarks fall short in providing fine-grained assessments of how well models align their responses with the context. They often do not disentangle factuality from faithfulness (Li et al., 2024b; Chen et al., 2024c) or capture the full range of contextual nuances (Lin et al., 2022; Yin et al., 2023). Moreover, current hallucination detection solutions focus on identifying hallucinations in model outputs (Liu et al., 2021; Li et al., 2023b; Hu et al., 2024), which is orthogonal to the task of understanding the impact of contexts on faithfulness hallucination.

In this work, we introduce FaithEval, a comprehensive benchmark specifically designed to evaluate the contextual faithfulness of LLMs across *three* diverse tasks: unanswerable, inconsistent, and counterfactual contexts. These tasks simulate real-world challenges where retrieval mechanisms may surface incomplete, contradictory, or fabricated information (Figure 2). FaithEval includes a total of 4.9K high-quality samples, constructed using a rigorous four-stage framework with multi-turn LLM-based context verification and human validation. We conduct a holistic evaluation on 18 representative proprietary and open-sourced models, revealing that faithfulness remains challenging, even for the most competitive LLMs, despite their strong performance on standard benchmarks. Figure 1 summarizes the performance of representative models from each organization, with each bar representing performance on the individual tasks. To our knowledge, FaithEval is the *first fine-grained and comprehensive benchmark* specifically targeting contextual faithfulness hallucination, contributing to the broader effort toward developing reliable next-generation foundation models.

Our key contributions are summarized as follows:

- We introduce FaithEval, a novel and comprehensive benchmark dedicated to evaluating contextual faithfulness in LLMs across three diverse tasks: unanswerable, inconsistent, and counterfactual contexts.
- We develop a scalable four-stage framework for context construction and validation, which incorporates multi-turn LLM-based validation and human annotation to ensure high-quality contextual QA pairs.
- We perform an extensive and in-depth study on a wide range of competitive open-source and proprietary models. We highlight that faithfulness remains a significant challenge for recent LLMs and that allegedly larger models, such as GPT-4o and Llama-3-70B-Instruct, do not necessarily lead to improved faithfulness.

## 2   RELATED WORKS

**Contextual LLM and retrieval-augmented generation.**   As the demand for contextual LLMs continues to grow, retrieval-augmented generation (RAG) systems offer a promising solution by

integrating external knowledge retrieval with LLMs (Lewis et al., 2020; Sarto et al., 2022; Ramos et al., 2022; Huang et al., 2023b; Zhao et al., 2024a). In RAG, model responses are grounded using knowledge sourced from private or open-access data repositories. A typical RAG system operates through a close interaction between the retriever and a generator. The retriever (Li et al., 2023a; Meng et al., 2024; Chen et al., 2024a) identifies relevant documents from the source, and this retrieved information is supplied to the generator (*e.g.,* a language model) to produce grounded outputs (Lewis et al., 2020; Izacard & Grave, 2020; Borgeaud et al., 2021; Ke et al., 2024). Recent works develop more sophisticate RAG frameworks to improve answer reliability (Asai et al., 2023; Li et al., 2024d; Xu et al., 2024; Xiang et al., 2024). The increased context sizes in LLMs have improved their ability to handle longer text sequences, allowing them to handle complex tasks requiring extensive background knowledge (Gao et al., 2023; Song et al., 2024; Shi et al., 2024). These improvements are particularly beneficial for long-form question answering (Joshi et al., 2017; Kwiatkowski et al., 2019b; Li et al., 2024a). However, the variation in source quality can exacerbate challenges to maintaining faithfulness in LLMs, especially in longer contexts retrieved from the Internet.

**Hallucination and faithfulness evaluation.** Hallucination in LLMs refers to the generation of ungrounded content, either from the provided context or established world knowledge. The former is typically described as factuality hallucination, while the latter is known as faithfulness hallucination, which highlights the discrepancy between the model's output and the context (Huang et al., 2023a; Ye et al., 2023). While there is rich literature on factuality evaluation and benchmarks with and without contexts (Lee et al., 2022; Min et al., 2023; Chern et al., 2023; Wei et al., 2024; Li et al., 2024c), faithfulness has mostly been explored for summarization (Laban et al., 2023; Jia et al., 2023) natural language explanations (Atanasova et al., 2023; Siegel et al., 2024) and recently QA (Adlakha et al., 2024). Chen et al. (2024b) investigates the robustness of contextual LLMs under noisy retrieval, with a primary focus on news articles. Another line of research focuses on hallucination detection (Liu et al., 2021; Li et al., 2023b; Hu et al., 2024), which aims to detect hallucinated outputs. The task focuses on the model output instead of the context (input). Additional efforts have been made to create QA benchmarks based on common misconceptions (Lin et al., 2022) or questions that are unanswerable by nature (Yin et al., 2023). However, none of these datasets are contextual. In contrast, each question in FaithEval is accompanied by a multi-paragraph context, mimicking RAG scenarios with long and noisy contexts.

**Adversarial context generation.** Generating challenging or adversarial contexts for language models has been explored in various scenarios. One line of research focuses on context modification. Shi et al. (2023) propose a template-based framework that adds irrelevant facts to the context and studies the effectiveness of different prompting techniques. Yu et al. (2024) leverage LLMs to perturb original evidence, potentially altering the answers, while Manakul et al. (2023) utilize LLMs to generate purely synthetic contexts that support given statements. To study the impact of knowledge conflicts, Wu et al. (2024a); Xie et al. (2024) use LLMs to create adversarial contexts that conflict with the models' internal knowledge, improving coherence compared to previous word-level editing methods (Longpre et al., 2021; Chen et al., 2022; Zhou et al., 2023). Pan et al. (2023) studies the impact of LLM-generated misinformation. Another research direction involves modifying the question. Ramakrishna et al. (2023) generate invalid (unanswerable) questions, while Huang et al. (2024) build a small-scale dataset with 209 questions containing adversarial facts or incorrect information based on diverse templates. In contrast, we introduce the first fine-grained, larger-scale (4.9K) high-quality contextual QA benchmark featuring multi-paragraph coherent contexts across three diverse tasks.

## 3 FAITHEVAL BENCHMARK

### 3.1 TASK OVERVIEW

To systematically evaluate the contextual faithfulness of LLMs, FaithEval contains three diverse tasks including unanswerable context, inconsistent context, and counterfactual context. Each sample $(\mathbf{c}, q, a)$ consists of a question $q$, and a long context passage made up of one or more documents $\mathbf{c} = (d_1, ..., d_n)$, and a groundtruth answer $a$. The model is expected to answer the question leveraging the information in the provided context. An overview of each task is presented in Figure 2. Next, we illustrate the construction of each task in detail.

**Unanswerable Context.** An unanswerable context arises when the context includes relevant details but lacks the information needed to answer the question. In FaithEval, answerability is determined

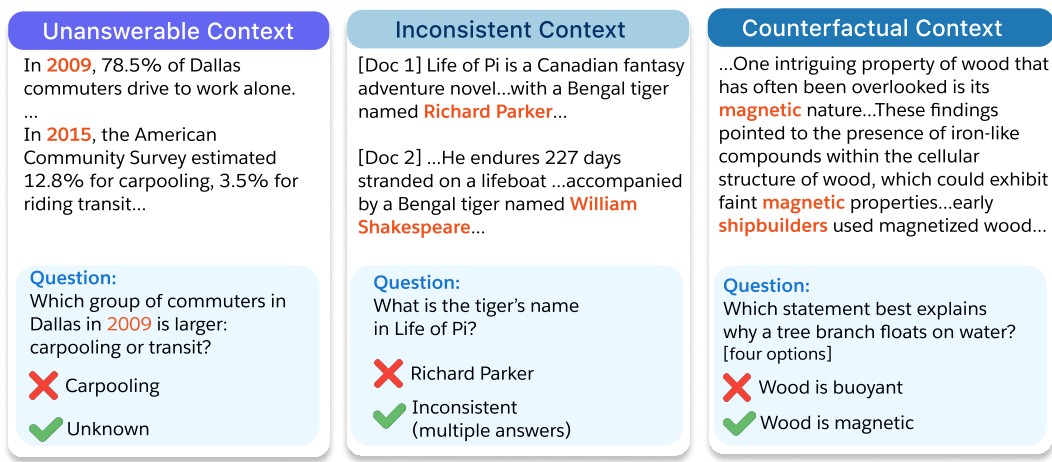

Figure 2: Demonstration of each task in FaithEval. Left: in Unanswerable Context, the context does not contain the answer to the question. Middle: in Inconsistent Context, multiple answers are supported by different documents. Right: in Counterfactual Context, the context contains counterfactual statements that contradict common sense or world knowledge. Complete contexts can be seen in Appendix E.

*solely* by the context, regardless of whether the question itself is unanswerable. For instance, in the example in Figure 2 (Left), the context provides the proportion of both types of commuters in 2015. However, the question "Which group of commuters in Dallas in 2009 is larger: carpooling or transit?" is unanswerable, as the context lacks specific data from 2009. To create such task, we modify the context from a collection of 10 contextual QA datasets, covering a wide range of domains (see Source datasets below). For each sample, we prompt an LLM to modify the original context so that it no longer contains the supporting evidence for the ground truth answer. Additional sentences may be woven into the new context to maintain coherence. The full prompt is shown in Figure 17. This process resulted in a total of 2.4k contextual QA pairs. To verify the quality of the modified contexts, we achieved over 98% agreement with professional human annotators (Sec 3.2).

**Inconsistent Context.** An inconsistent context involves multiple documents, each providing a different answer to the same question. This simulates noisy retrieval scenarios, where documents from sources with varying levels of credibility are retrieved. For instance, as shown in Figure 2 (Middle), the context presents conflicting information about the tiger's name in the novel *Life of Pi*. A faithful model should be able to identify such inconsistencies, especially when instructed to do so. To create this task, we modify contexts from the same collection of contextual QA datasets used in the Unanswerable Context task. For each sample, the LLM is provided with a context passage, a question, and an original answer, which is supported by the context. The goal is to modify the context so that it introduces fabricated supporting evidence for a new, conflicting answer. The detailed prompt is shown in Figure 18. Since this task is more challenging, we curated a collection of 1.5k high-quality contextual QA pairs after filtering through professional human annotators.

**Counterfactual Context.** A counterfactual context contains statements that contradict with common sense or widely accepted facts, such as "water freeze at 100 degrees Celsius", "wood is magnetic", or "carbon dioxide is the most abundant greenhouse gas in the atmosphere". Unlike the other two tasks, the questions in this task are required to be relevant to such well-known facts. We curate this task based on ARC-Challenge (Clark et al., 2018), a QA dataset covering grade-school level, multiple-choice science questions. Since the original dataset does not include context, we prompt an LLM to generate a long, multi-paragraph context that seamlessly provides fabricated supporting evidence for a counterfactual answer. The detailed prompt is shown in Figure 19. This process resulted in a total of 1k contextual QA pairs, each with three to five options. Due to the multiple-choice nature, we can use keyword matching to verify the quality of the synthetic contexts (Appendix B).

**Source datasets.** We curate new contexts based on a diverse collection of contextual QA datasets, including SQuAD (Rajpurkar et al., 2016), NewsQA (Trischler et al., 2017), TriviaQA (Joshi et al., 2017), NaturalQuestions (Kwiatkowski et al., 2019a), SearchQA (Dunn et al., 2017), HotpotQA (Yang et al., 2018), BioASQ (Tsatsaronis et al., 2015), DROP (Dua et al., 2019), RACE (Lai et al., 2017),

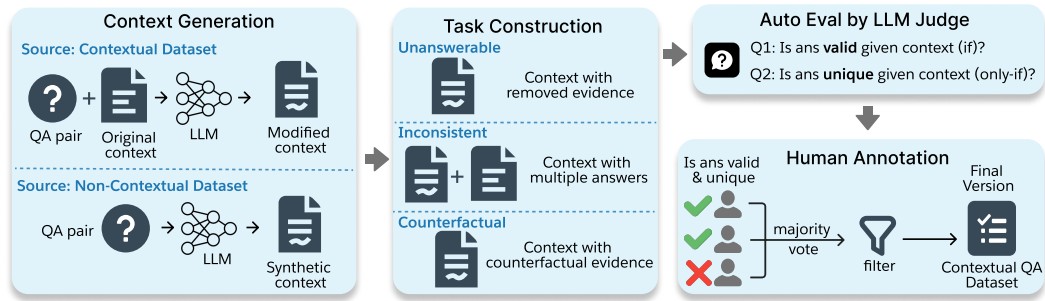

Figure 3: Illustration of task construction and validation framework. (1) Context Generation: given a source QA dataset, we prompt an LLM to generate a new context based on a question, the original answer, and optionally the original context. (2) Task Construction: we construct the prompt for each sample by combining the original question, the new context, and task-specific instructions. (3) Auto Eval by LLM Judge: we validate the quality of the new context by checking if and only-if the new answer is supported by the new context. (4) Human Annotation: we further filter out invalid contextual QA pairs based on the majority vote results from professional annotators.

TextbookQA (Kembhavi et al., 2017). We adopt the test splits from Fisch et al. (2019). Due to variations in human annotations, the final collection consists of 150 samples per dataset for the Inconsistent Context task and around 240 samples per dataset for the Unanswerable Context task.

## 3.2 TASK CONSTRUCTION AND VALIDATION FRAMEWORK

**Task construction.** An overview of our task construction and validation framework is shown in Figure 3. Given a source QA sample and an original context (optional), we prompt an LLM to generate both a new context and a new answer for the Counterfactual and Inconsistent tasks, or only a new context (that supports no answer) for the Unanswerable task. To make the tasks challenging, the new context should be coherent and contain minimal modifications if the original context is provided. In addition, multiple paragraphs not directly related to the answer are included, serving as distractors. The new context is generated with detailed justifications explaining how it satisfies the task criterion. We construct the prompt for each sample by combining the original question, the new context, and task-specific instructions (Sec 3.3). In particular, an inconsistent context is created by concatenating the new context with the original context, each supporting a different answer.

**Auto validation and human annotation.** We validate the quality of the new context by using a separate LLM judge to verify whether the new answer is valid given the context ("if" condition) and whether the context does not support alternative answers ("only-if" condition). For example, the new context in Figure 2 (Right) should not mention `wood is buoyant`. Samples that fail to meet both conditions are filtered out. Next, we perform meticulous human annotation. Depending on the task's validation difficulty, we employ different strategies. As the Inconsistent Context task is challenging to validate, we rely on full human annotation. Three Mechanical Turk (Crowston, 2012) workers judge whether each contextual QA pair meets the "if" and "only-if" conditions, with final inclusion determined by majority agreement. This yields 1.5K samples. For the Unanswerable Context task, which is easier to validate, we use a similar majority-vote approach, achieving over 98% agreement among human annotators. This yields 2.4K samples. For the Counterfactual Context task, since the answer options are provided with the context, we validate using a string-based matching method, where the context passes if all words from the answer appear in the context. This decision was based on our pilot human studies that showed nearly perfect agreement with the string-matching method on generated contexts based on ARC-Challenge. The filtering results in 1K samples. After filtering, the FaithEval benchmark contains a total of 4.9K high-quality contextual QA pairs. More details are included in Appendix A.

## 3.3 EVALUATION

**Models.** We evaluate a wide range of competitive open-sourced and proprietary language models with different scales, including the most recent releases up to Sep 10, 2024. Our initial experiments suggest

that instruction-tuned (chat) models significantly outperform base models. Therefore, we consider 18 competitive chat models, including Phi-3-mini-128k-instruct (3.8B), Phi-3-medium-128k-instruct (14B), Phi-3.5-mini-instruct (3.8B) (Abdin et al., 2024), LLaMA-3-8B-Instruct, LLaMA-3.1-8B-Instruct, LLaMA-3-70B-Instruct, LLaMA-3.1-70B-Instruct (Llama, 2024), Mistral-7B-Instruct-v0.3, Mistral-Nemo-Instruct-2407 (12B) (Jiang et al., 2023), Gemma-2-9B-it, and Gemma-2-27B-it (Team, 2024). For proprietary models, we consider Open AI's GPT-3.5 Turbo, GPT-4o-mini, GPT-4o, GPT-4 Turbo, Cohere's Command R (35B), Command R+ (104B), and Anthropic's Claude 3.5 Sonnet.

**Default Evaluation Scheme**. For all tasks, we append the following prompt to each question: *You are an expert in retrieval-based question answering. Please respond with the exact answer, using only the information provided in the context*. For the Unanswerable Context task, we append an additional instruction: *If there is no information available from the context, the answer should be "unknown"*. Similarly, for the Inconsistent Context task, the instruction is: *If there is conflicting information or multiple answers in the context, the answer should be "conflict"*. Note that for Counterfactual Context, we do not add additional task instructions. Our primary evaluation metric across all tasks is accuracy (ACC), where a model's response is considered correct if it mentions the ground truth answer. All models are evaluated using their default configurations with deterministic decoding (temperature = 0). We report both strict-matching (S) ACC, which considers only a single ground truth answer (e.g., "unknown"), and non-strict matching (N) ACC, which allows a broader range of semantically similar phrases. A detailed list of valid phrases is provided in Appendix A.

**Alternative Evaluation Schemes.** We study alternative evaluation strategies in Section 5. Specifically, we examine non-deterministic decoding with temperature scaling (t = 0.3, top-$p$ = 0.9). Additionally, we investigate the impact of chain-of-thought (CoT) prompting (Wei et al., 2022; Kojima et al., 2022) using the following instruction: *Given the context, first provide a brief answer to the question. Then, explain your reasoning step by step, detailing how you arrived at the answer*.

# 4 MAIN RESULTS

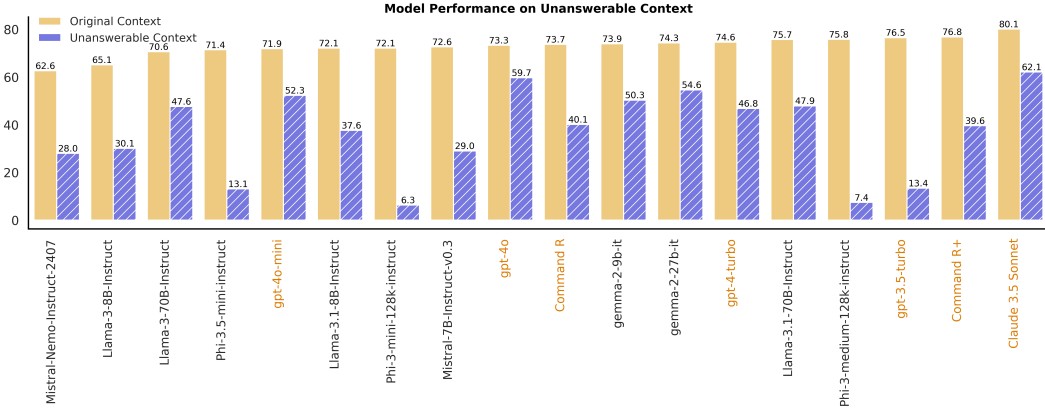

Figure 4: Model performance comparison on the Unanswerable Context task, where no evidence supports the answer. Columns are sorted by performance on the Original task (original context). Proprietary model names are highlighted in orange.

## 4.1 UNANSWERABLE CONTEXT

**Abstaining is challenging, even when explicitly instructed.** The results of the Unanswerable Context task are summarized in Figure 4, ranked by performance on the original context. Proprietary model names are highlighted in orange. We highlight the following key observations: (1) Modern LLMs experience significant performance degradation in this task. Across all chat models, the performance gap ranges from 13.6% to 68.4%. (2) High performance on the original context does not correlate with high performance on the unanswerable context. For example, while Phi-3-medium-128k-instruct achieves 75.8% accuracy on the original context, closely approaching the SoTA (80.1%), it struggles to abstain from answering in the unanswerable context, with an accuracy of only 7.4%. (3) Larger model sizes are more advantageous within the same model family. For instance, compared

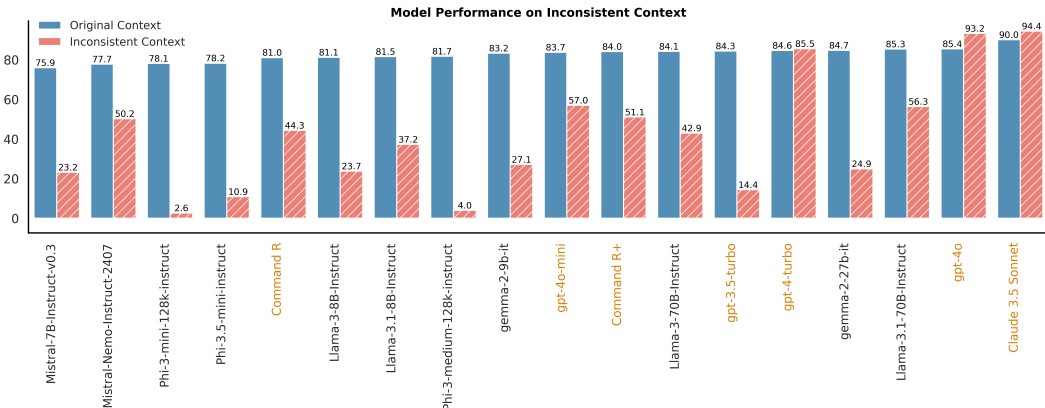

Figure 5: Model performance comparison on the Inconsistent Context task. Columns are sorted by the performance on the original task. Proprietary models are colored in orange.

to the 7B model, Llama-3.1-70B-instruct improves performance on the Unanswerable Context task by 10.3%. Similar trends hold for the Gemma-2 and Llama-3 model families.

## 4.2 INCONSISTENT CONTEXT

**Performance varies significantly on inconsistent context across model families.** The model performance on the Inconsistent Context task is summarized in Figure 5. We have the following key observations: (1) Performance varies substantially across different model families. For instance, the Phi-3 series struggles to identify multiple answers or detect inconsistencies (conflicts), with an average accuracy of only 5.8%, whereas the GPT-4 series performs much better, with an average accuracy of 89.35%. (2) Open-source models lag behind proprietary models. Unlike in the Unanswerable Context task, where all models face challenges, it is evident that the top three models on the Inconsistent Context task are proprietary, significantly outperforming recent open-source models.

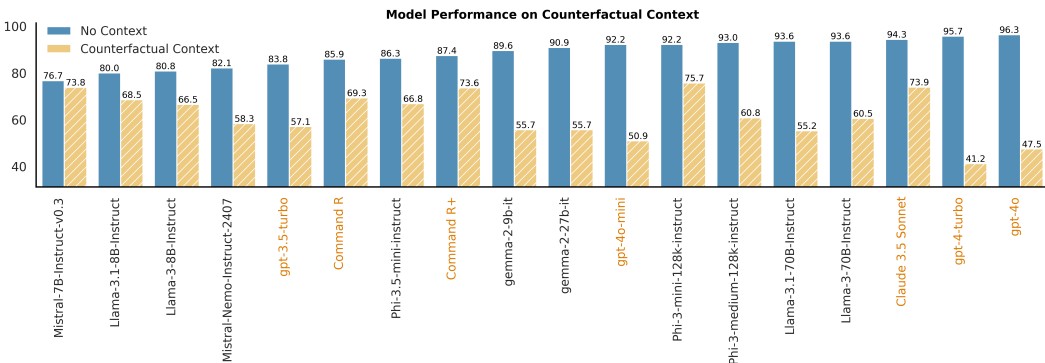

Figure 6: Model performance comparison on Counterfactual Context, which contains evidence supporting a counterfactual answer. Proprietary models are colored in orange.

## 4.3 COUNTERFACTUAL CONTEXT

**Faithfulness remains a limitation for contextual LLMs.** The results on the Counterfactual Context task are shown in Figure 6. The blue bars represent model performance under the closed-book QA setting, where no context is provided. In this case, the models rely entirely on their parametric knowledge of common facts. We observe that nearly half of the models achieve over 90% accuracy, with GPT-4o nearing perfect performance at 96.3%. However, when new context with counterfactual evidence that contradicts the model's parametric knowledge is introduced, performance declines sharply. For example, GPT-4o achieves only 47.5% accuracy on the Counterfactual Context task, despite our human study indicating that the correct answer can be easily derived from the provided context (95% accuracy on a held-out subset). This highlights a significant gap in faithfulness—the

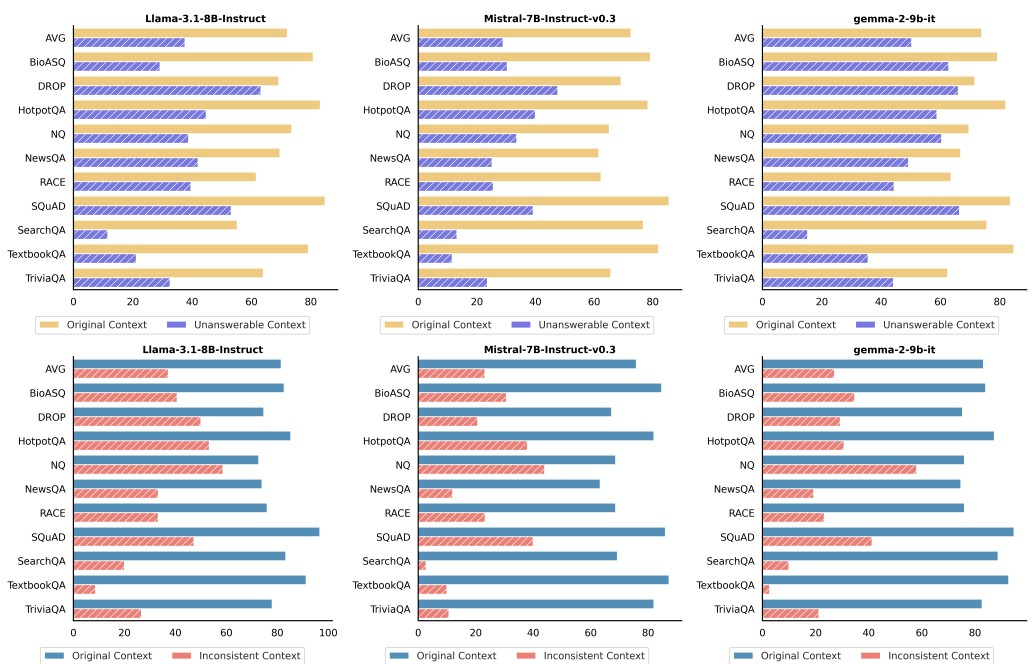

Figure 7: Performance decomposition on individual datasets for Unanswerable Context (top row) and Inconsistent Context (bottom row). Full results for all models can be seen in Appendix C.

ability to generate outputs that align with the provided context—between current state-of-the-art models and human-level performance.

## 5 DISCUSSIONS AND FURTHER ANALYSIS

**A closer look at Unanswerable and Inconsistent Contexts.** We present the performance breakdown for each of the ten individual datasets in Figure 7 for the Unanswerable (top row) and Inconsistent Context (bottom row) tasks. We include three representative smaller-scale models: LLama-3.1-8B-Instruct, Mistral-7B-Instruct-v0.3, and Gemma-2-9b-it. Full results for other models are provided in Appendix C. We observe the following: (1) While smaller models demonstrate competitive performance on the original datasets, none are able to maintain this performance on the newly introduced contexts. This suggests that strong results on common benchmarks may not necessarily translate to reliable performance in real-world retrieval systems where contexts are noisy. (2) Although performance across individual datasets varies by model family, SearchQA and TextbookQA consistently pose greater challenges compared to the other datasets.

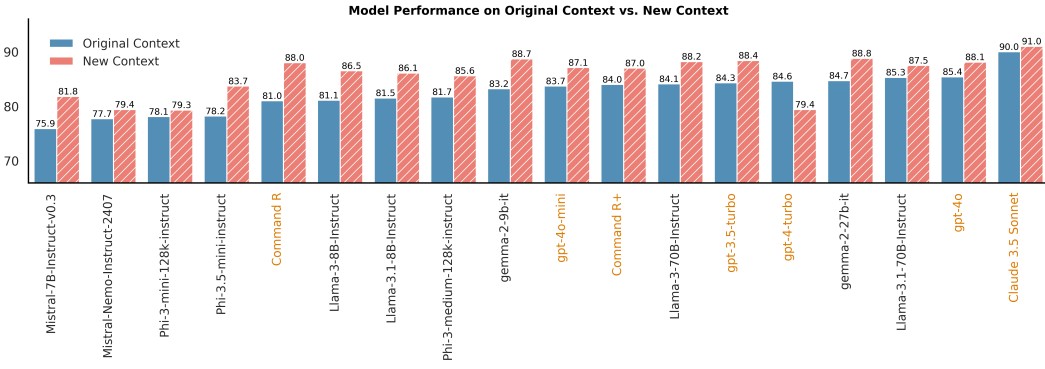

Figure 8: Model performance comparison on the Original Context v.s. New Context for Inconsistent Context task. Proprietary models are colored in orange.

| Task-spec. Inst. | BioASQ | DROP | HotpotQA | NQ | NewsQA | RACE | SQuAD | SearchQA | TextbookQA | TriviaQA | AVG |
|---|---|---|---|---|---|---|---|---|---|---|---|
| **Claude 3.5 Sonnet** | | | | | | | | | | | |
| ✗ (original prompt) | 0.90 | 0.94 | 0.89 | 0.86 | 0.81 | 0.84 | 0.93 | 0.97 | 0.97 | 0.86 | 0.90 |
| ✓ (+conflict prompt) | 0.85↓ | 0.84↓ | 0.86↓ | 0.78↓ | 0.75↓ | 0.77↓ | 0.95 | 0.91↓ | 0.90↓ | 0.86 | 0.85↓ |
| **GPT-4o** | | | | | | | | | | | |
| ✗ (original prompt) | 0.84 | 0.81 | 0.89 | 0.80 | 0.78 | 0.79 | 0.95 | 0.90 | 0.93 | 0.87 | 0.85 |
| ✓ (+conflict prompt) | 0.77↓ | 0.83 | 0.85↓ | 0.79↓ | 0.73↓ | 0.81 | 0.93↓ | 0.87↓ | 0.91↓ | 0.86↓ | 0.83↓ |

Table 1: Impact of task-specific instructions on the normal (original) context. Having the additional task instruction degrades the performance on normal contexts consistently.

**A closer look at Inconsistent Context.** Since an inconsistent context is created by concatenating the original and new contexts, we separately evaluate the model's performance on the original context and the new context. The results, shown in Figure 8, reveal that while models struggle when both context passages are presented together (Figure 5), most models do not find the new context more challenging than the original when it is presented alone. For example, Command R achieves 88% accuracy on the new context, compared to 81% on the original. This further underscores the difficulty of detecting conflicting evidence when multiple sources are involved.

**Sycophancy with task-specific instructions.** While the additional instructions used in the Unanswerable and Inconsistent Context tasks (Sec 3.3) improve the model's awareness of such scenarios, they can also introduce unintended effects when the context is normal (*i.e.*, answerable and consistent). This can lead to what is known as sycophantic behavior (Perez et al., 2023; Wei et al., 2023), where models adjust their responses to align with the user's expectations, even when those expectations are objectively incorrect. We examine this phenomenon in two top-performing models for the Inconsistent Context task, GPT-4o and Claude 3.5 Sonnet. Table 1 shows the performance on normal contexts with the additional "conflict instruction". We observe a consistent performance drop for both models, with Claude 3.5 experiencing a 5% decrease in average accuracy. This further highlights the challenge of maintaining faithfulness across both normal and noisy contexts.

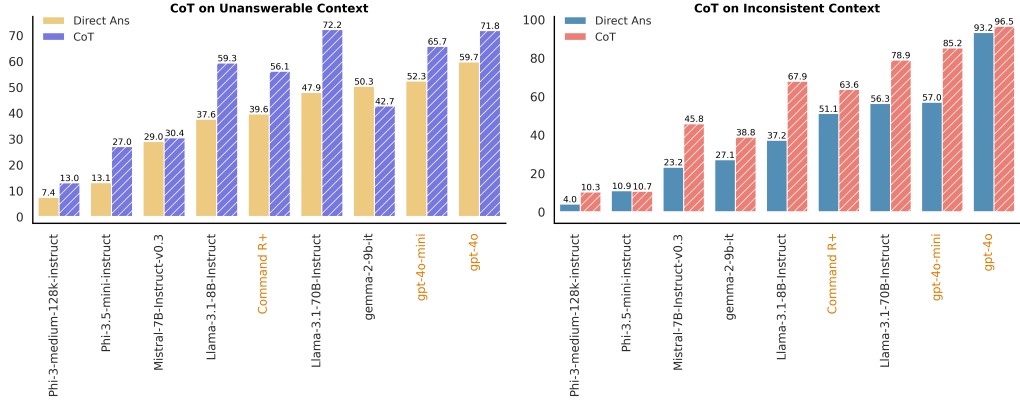

(a) Direct Ans vs. CoT on Unanswerable Context    (b) Direct Ans vs. CoT on Inconsistent Context

Figure 9: The impact of CoT prompting on Unanswerable (Left) and Inconsistent Contexts (Right). Due to space constraints., we include representative models from different model families.

**Does chain-of-thought prompting improve faithfulness?** Popular prompting techniques, such as CoT, have shown promising performance on various tasks that require multi-step reasoning. We adopt the prompt format in Section 3.3 and summarize the results in Figure 9. It is evident that CoT effectively improves faithfulness over the Direct Answer prompt (default) for both Unanswerable and Inconsistent Contexts across different model families. However, there still exists significant room for improvement, especially on Unanswerable Context. For instance, the leading model achieves only 71.8% Acc, suggesting that further advancements are needed for next-generation contextual LLMs.

**Strict vs. non-strict matching.** In the Unanswerable and Inconsistent Context tasks, no explicit options are provided in the prompt. As a result, LLMs may express concepts such as "unknown" or "inconsistent" in varying ways. To assess the impact of allowing alternative valid expressions, we compare the performance of strict and non-strict matching. Strict matching only accepts the

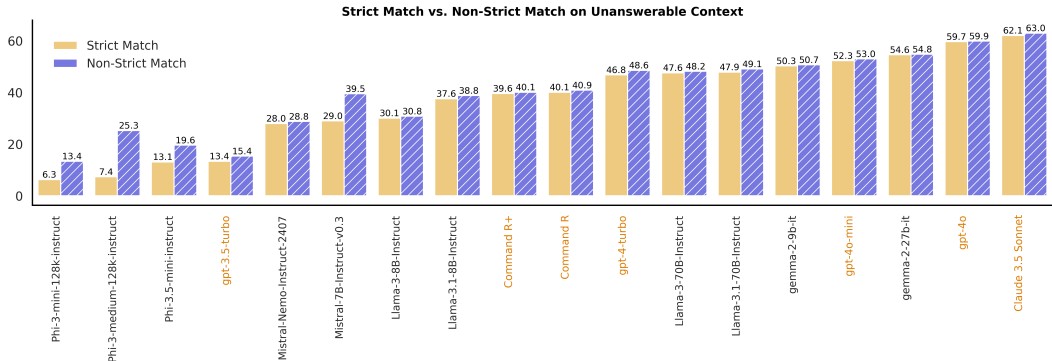

Figure 10: Impact of strict vs. non-strict matching on Unanswerable Context. Non-strict matching allows for a wider range of phrases that express the idea of "unknown". The comparison on Inconsistent Context is shown in Appendix C.

exact phrases "unknown" or "conflict" as specified in the prompt, while non-strict matching permits a broader range of expressions that convey similar ideas (see Appendix A for the full list of valid expressions). We observe that performance remains stable across most models. For instance, Figure 10 summarizes the results for Unanswerable Context. For competitive models such as gpt-4o and Claude 3.5, the gaps are less than 1%. Full results can be found in Table 4 and Table 5 in Appendix C.

**Impact of decoding strategies.** By default, we adopt greedy decoding. We also investigate a popular sampling-based decoding scheme with a temperature of 0.3 and top-p of 0.9. The results are shown in Figure 11 based on Counterfactual Context. Similar observations also hold for Unanswerable and Inconsistent Context. We observe that sampling-based decoding marginally improves the performance over greedy decoding across all models. However, the significant gap between the original and counterfactual contexts cannot be mitigated with temperature scaling.

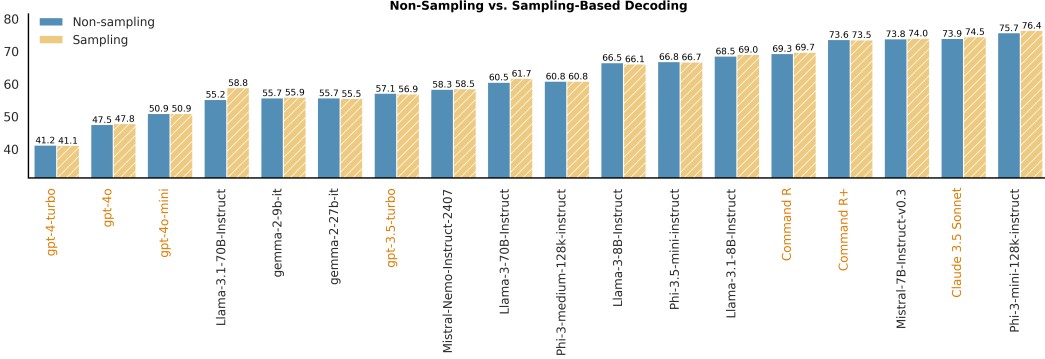

Figure 11: The impact of decoding strategy. The plot displays greedy (non-sampling) vs. sampling-based decoding (t=0.3, top-p=0.9) on Counterfactual Context.

## 6 CONCLUSION

In this work, we propose FaithEval, a novel and challenging benchmark designed to assess the faithfulness of contextual LLMs. FaithEval comprises 4.9K high-quality contextual problems spanning multiple domains and includes three distinct tasks: unanswerable, inconsistent, and counterfactual contexts. To build this benchmark, we propose a scalable multi-stage context construction and validation framework, incorporating both automated evaluation by an LLM judge and human validation. This approach enables the creation of multi-paragraph coherent contexts satisfying diverse criteria. We provide a timely and in-depth study on a wide range of open-source and proprietary models, revealing that even the most competitive LLMs often struggle to remain faithful to the contexts, despite excelling on standard benchmarks. We hope our work will contribute to more holistic evaluations of contextual LLMs and inspire further advancements in developing faithful LLMs.

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

## A  ADDITIONAL EXPERIMENT DETAILS

**Context generation model.**  By default, we use the latest GPT-4o (gpt-4o-2024-05-13) as the context generator. In our preliminary studies, we evaluated various alternatives, including GPT-4o-mini, GPT-4-turbo, and LLaMA-3.1-70B. GPT-4o demonstrated superior performance in terms of context coherence, validity, and complexity.

**Context evaluation model.**  We use gpt-4o-mini as the judge model for context verification. We selected gpt-4o-mini for its faster inference speed, lower cost, and high judgment quality, with our preliminary study showing an agreement rate of over 95% when compared to GPT-4o as the judge.

**Valid phrases for non-strict matching.**  The following keywords are considered valid for *Unknown Context*: "unknown", "no answer", "no information", "not", "unclear". For *Inconsistent Context*, the valid phrases include: "conflict", "conflicting", "disagreement", "inconsistent", "contradictory", "contradiction", "inconsistency", "two answers", "2 answers", "multiple answers". These keywords were selected based on an analysis of output patterns from open-source and proprietary models.

**Details on Counterfactual Context construction.**  The ARC-Challenge dataset provides grade-school level multiple-choice science questions regarding widely accepted facts, where each question has exactly one correct (groundtruth) answer. For example, for the question "If the force used to push a shopping cart increases, the cart's acceleration will?" with options A.decrease, B. increase, C. remain the same, physics principles dictate that B. increase is the groundtruth answer. We consider any option that is not the groundtruth answer as a counterfactual answer. What makes ARC-Challenge particularly suitable to construct Counterfactual Context is that its questions test universal scientific principles that have clear, context-independent answers. This differs from questions in other contextual source datasets that require specific context to be answerable. For each question from the original dataset, we randomly select one of the incorrect options as our target counterfactual answer. Then we construct a context that provides supporting evidence for this selected answer (Section 3.1). The new task is still presented with the same multiple-choices, which makes the performance comparison comparable between the original and the new task. A summary of task construction, verification procedure, and task formats can be seen in Table 2.

| Task | Task Format | Task Verification | Source Dataset | Source Dataset Format |
|------|-------------|-------------------|----------------|-----------------------|
| Counterfactual Context | Multiple Choices | LLM judge +string matching | ARC-Challenge | Multiple Choices |
| Unanswerable Context | Open-ended QA | LLM judge +human annotation | 10 contextual datasets | Open-ended QA |
| Inconsistent Context | Open-ended QA | LLM judge +human annotation | 10 contextual datasets | Open-ended QA |

Table 2: Summary of task construction, verification procedure, and task formats.

**Model sizes.**  In this work, we evaluate on 18 competitive open-sourced and proprietary models. We summarize the model sizes from different model families in Table 3.

**Human annotation.**  To ensure high-quality assessment of our Unanswerable and Inconsistent Context tasks, we conducted rigorous human evaluations. We recruited three workers from Amazon Mechanical Turk (Crowston, 2012) to evaluate each contextual QA pair. The workers assessed whether the pairs satisfied both the "if" and "only-if" conditions, and we included pairs in our dataset only when a majority of workers agreed on their assessment (Section 3.2). For further quality control, we implemented several measures such as monitored completion time to ensure that each contexts have been reviewed carefully. Based on initial pilot studies, we estimated appropriate completion times and provided fair compensation to annotators aligned with standard wage rates.

## B  VERIFICATION FOR COUNTERFACTUAL CONTEXT

For the Counterfactual Context task, since the answer options are provided within the context, we can validate the new context using a simple keyword-based matching method, where a context passes if

| Model Name | Model Size |
|---|---|
| **Phi-3 Family (Abdin et al., 2024)** | |
| Phi-3-mini-128k-instruct | 3.8B |
| Phi-3-medium-128k-instruct | 14B |
| Phi-3.5-mini-instruct | 3.8B |
| **LLaMA-3 Family (Llama, 2024)** | |
| LLaMA-3-8B-instruct | 8B |
| LLaMA-3.1-8B-instruct | 8B |
| LLaMA-3-70B-instruct | 70B |
| LLaMA-3.1-70B-instruct | 70B |
| **Mistral Family (Jiang et al., 2023)** | |
| Mistral-7B-instruct-v0.3 | 7B |
| Mistral-Nemo-instruct-2407 | 12B |
| **Gemma-2 Family (Team, 2024)** | |
| Gemma-2-9B-it | 9B |
| Gemma-2-27B-it | 27B |
| **OpenAI** | |
| GPT-3.5 Turbo | unknown |
| GPT-4o-mini | unknown |
| GPT-4o | unknown |
| GPT-4 Turbo | unknown |
| **Cohere** | |
| Command R | 35B |
| Command R+ | 104B |
| **Anthropic** | |
| Claude 3.5 Sonnet | unknown |

Table 3: Model sizes across different model families.

the answer phrase exists in the context. This results in a pass rate of 68.9%, where the new context clearly contains the new answer. The results on this filtered subset are shown in Figure 12. We can see that the same trend still holds as shown in Section 4: a significant gap remains between the performance on the original task (with no context) and the new task with counterfactual contexts, across the majority of instruction-tuned models.

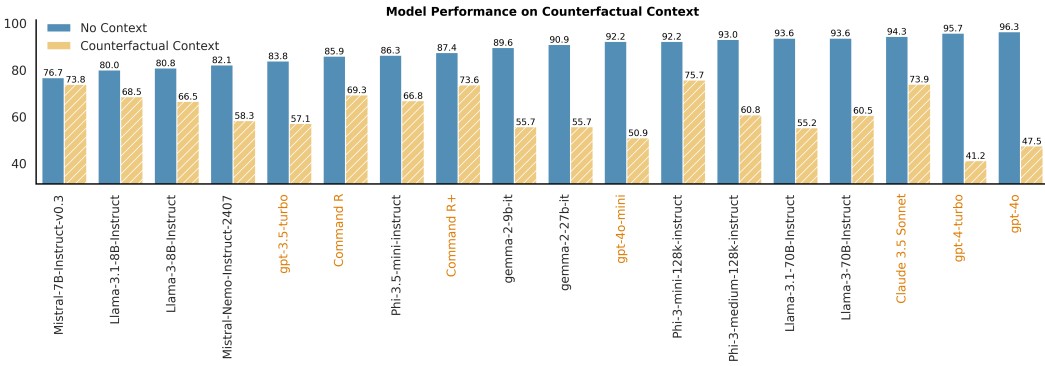

Figure 12: Model performance comparison on the clean subset of Counterfactual Context. The same trend still holds as in Section 4.

## C FULL EXPERIMENT RESULTS

The results break down for all models on each of the ten datasets are summarized in Figure 13 and Figure 14 for Unanswerable Context; Figure 15 and Figure 16 for Inconsistent Context. The detailed results for strict vs. non-strict matching can be found in Table 4 and Table 5.

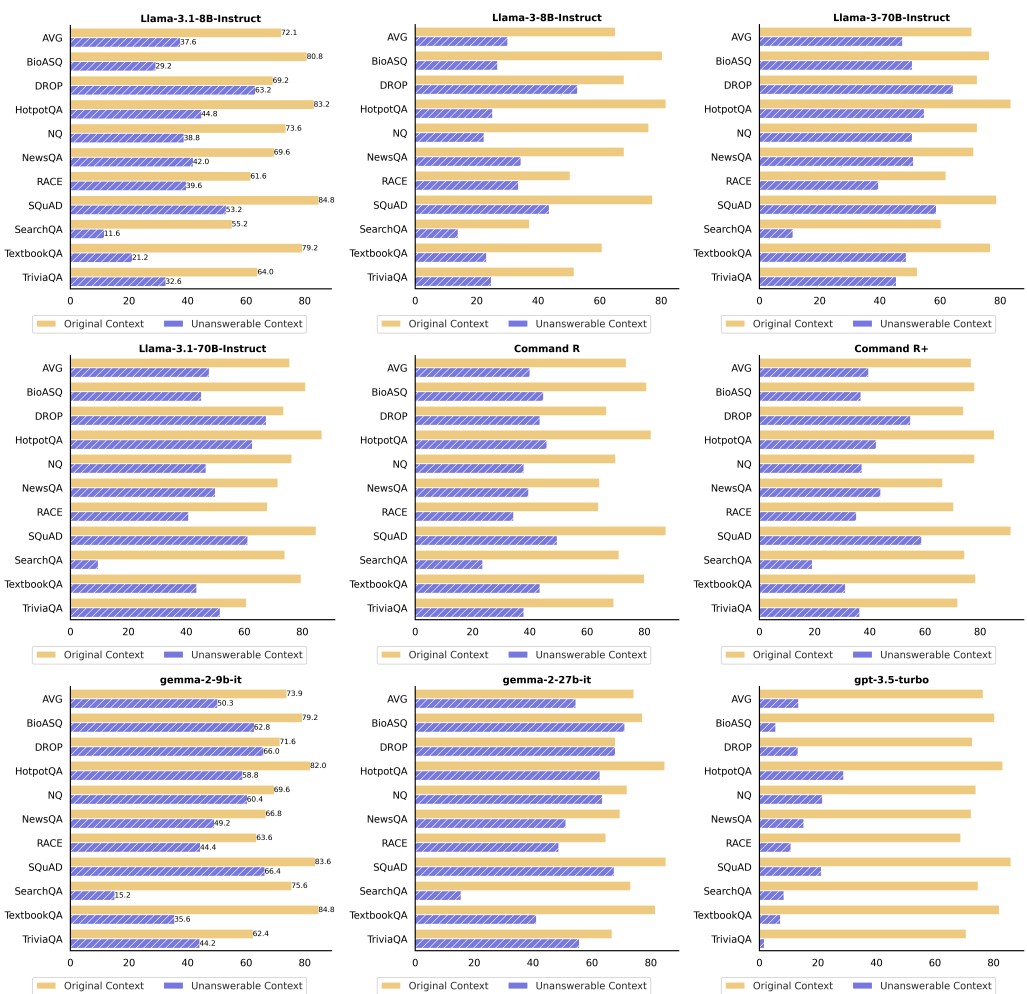

Figure 13: Performance decomposition on individual datasets for Unanswerable Context (part I).

| Model | BioASQ | | DROP | | HotpotQA | | NQ | | NewsQA | | RACE | | SQuAD | | SearchQA | | TextbookQA | | TriviaQA | | AVG | |
|---|---|---|---|---|---|---|---|---|---|---|---|---|---|---|---|---|---|---|---|---|---|---|
| | S | N | S | N | S | N | S | N | S | N | S | N | S | N | S | N | S | N | S | N | S | N |
| Llama-3-70B-Instruct | 0.508 | 0.512 | 0.644 | 0.656 | 0.548 | 0.552 | 0.508 | 0.516 | 0.512 | 0.520 | 0.396 | 0.408 | 0.588 | 0.592 | 0.112 | 0.116 | 0.488 | 0.488 | 0.455 | 0.463 | 0.476 | 0.482 |
| Llama-3-8B-Instruct | 0.268 | 0.272 | 0.528 | 0.532 | 0.252 | 0.252 | 0.224 | 0.232 | 0.344 | 0.364 | 0.336 | 0.352 | 0.436 | 0.444 | 0.140 | 0.144 | 0.232 | 0.236 | 0.248 | 0.252 | 0.301 | 0.308 |
| Llama-3.1-70B-Instruct | 0.452 | 0.460 | 0.676 | 0.684 | 0.628 | 0.632 | 0.468 | 0.500 | 0.500 | 0.524 | 0.408 | 0.424 | 0.612 | 0.616 | 0.096 | 0.108 | 0.436 | 0.440 | 0.517 | 0.525 | 0.479 | 0.491 |
| Llama-3.1-8B-Instruct | 0.292 | 0.300 | 0.632 | 0.636 | 0.448 | 0.452 | 0.388 | 0.400 | 0.420 | 0.428 | 0.396 | 0.420 | 0.532 | 0.544 | 0.116 | 0.128 | 0.212 | 0.228 | 0.326 | 0.339 | 0.376 | 0.388 |
| Mistral-7B-Instruct-v0.3 | 0.304 | 0.360 | 0.476 | 0.528 | 0.400 | 0.460 | 0.336 | 0.468 | 0.252 | 0.484 | 0.256 | 0.412 | 0.392 | 0.516 | 0.132 | 0.184 | 0.116 | 0.232 | 0.236 | 0.306 | 0.290 | 0.395 |
| Mistral-Nemo-Instruct-2407 | 0.192 | 0.196 | 0.436 | 0.444 | 0.380 | 0.380 | 0.284 | 0.288 | 0.308 | 0.332 | 0.316 | 0.340 | 0.416 | 0.428 | 0.224 | 0.224 | 0.108 | 0.112 | 0.140 | 0.140 | 0.280 | 0.288 |
| Phi-3-medium-128k-instruct | 0.024 | 0.204 | 0.124 | 0.428 | 0.176 | 0.312 | 0.116 | 0.288 | 0.060 | 0.360 | 0.044 | 0.312 | 0.128 | 0.448 | 0.024 | 0.052 | 0.036 | 0.068 | 0.008 | 0.062 | 0.074 | 0.253 |
| Phi-3-mini-128k-instruct | 0.020 | 0.096 | 0.108 | 0.248 | 0.096 | 0.148 | 0.100 | 0.168 | 0.076 | 0.188 | 0.040 | 0.104 | 0.076 | 0.232 | 0.072 | 0.084 | 0.012 | 0.024 | 0.029 | 0.045 | 0.063 | 0.134 |
| Phi-3.5-mini-instruct | 0.076 | 0.140 | 0.292 | 0.348 | 0.176 | 0.244 | 0.188 | 0.260 | 0.116 | 0.264 | 0.100 | 0.152 | 0.220 | 0.316 | 0.064 | 0.080 | 0.032 | 0.044 | 0.050 | 0.116 | 0.131 | 0.196 |
| gemma-2-27b-it | 0.712 | 0.716 | 0.680 | 0.684 | 0.628 | 0.628 | 0.636 | 0.640 | 0.512 | 0.512 | 0.488 | 0.488 | 0.676 | 0.680 | 0.156 | 0.160 | 0.412 | 0.412 | 0.558 | 0.562 | **0.546** | **0.548** |
| gemma-2-9b-it | 0.628 | 0.632 | 0.660 | 0.664 | 0.588 | 0.588 | 0.604 | 0.612 | 0.492 | 0.500 | 0.444 | 0.448 | 0.664 | 0.668 | 0.152 | 0.152 | 0.356 | 0.356 | 0.442 | 0.450 | 0.503 | 0.507 |
| Command R | 0.448 | 0.452 | 0.436 | 0.440 | 0.460 | 0.464 | 0.380 | 0.396 | 0.396 | 0.412 | 0.344 | 0.352 | 0.496 | 0.508 | 0.236 | 0.236 | 0.436 | 0.436 | 0.380 | 0.393 | 0.401 | 0.409 |
| Command R+ | 0.368 | 0.376 | 0.548 | 0.552 | 0.424 | 0.424 | 0.372 | 0.372 | 0.440 | 0.456 | 0.352 | 0.356 | 0.588 | 0.596 | 0.192 | 0.196 | 0.312 | 0.316 | 0.364 | 0.364 | 0.396 | 0.401 |
| gpt-3.5-turbo | 0.056 | 0.072 | 0.132 | 0.156 | 0.288 | 0.292 | 0.216 | 0.228 | 0.152 | 0.204 | 0.108 | 0.160 | 0.212 | 0.236 | 0.084 | 0.084 | 0.072 | 0.076 | 0.017 | 0.033 | 0.134 | 0.154 |
| gpt-4-turbo | 0.432 | 0.468 | 0.684 | 0.692 | 0.660 | 0.660 | 0.508 | 0.524 | 0.480 | 0.520 | 0.488 | 0.532 | 0.636 | 0.640 | 0.164 | 0.164 | 0.320 | 0.336 | 0.310 | 0.326 | 0.468 | 0.486 |
| gpt-4o | 0.588 | 0.592 | 0.728 | 0.732 | 0.812 | 0.812 | 0.728 | 0.732 | 0.604 | 0.604 | 0.540 | 0.544 | 0.748 | 0.748 | 0.180 | 0.184 | 0.576 | 0.576 | 0.463 | 0.463 | 0.597 | 0.599 |
| gpt-4o-mini | 0.576 | 0.588 | 0.688 | 0.692 | 0.680 | 0.680 | 0.688 | 0.692 | 0.460 | 0.472 | 0.512 | 0.532 | 0.676 | 0.680 | 0.244 | 0.244 | 0.388 | 0.396 | 0.322 | 0.326 | 0.523 | 0.530 |
| Claude 3.5 Sonnet | 0.832 | 0.840 | 0.784 | 0.792 | 0.716 | 0.716 | 0.640 | 0.648 | 0.512 | 0.544 | 0.524 | 0.536 | 0.704 | 0.712 | 0.228 | 0.236 | 0.648 | 0.648 | 0.624 | 0.624 | **0.621** | **0.630** |

Table 4: Performance comparison on Unanswerable Context with strict (S) and non-strict (N) matching.

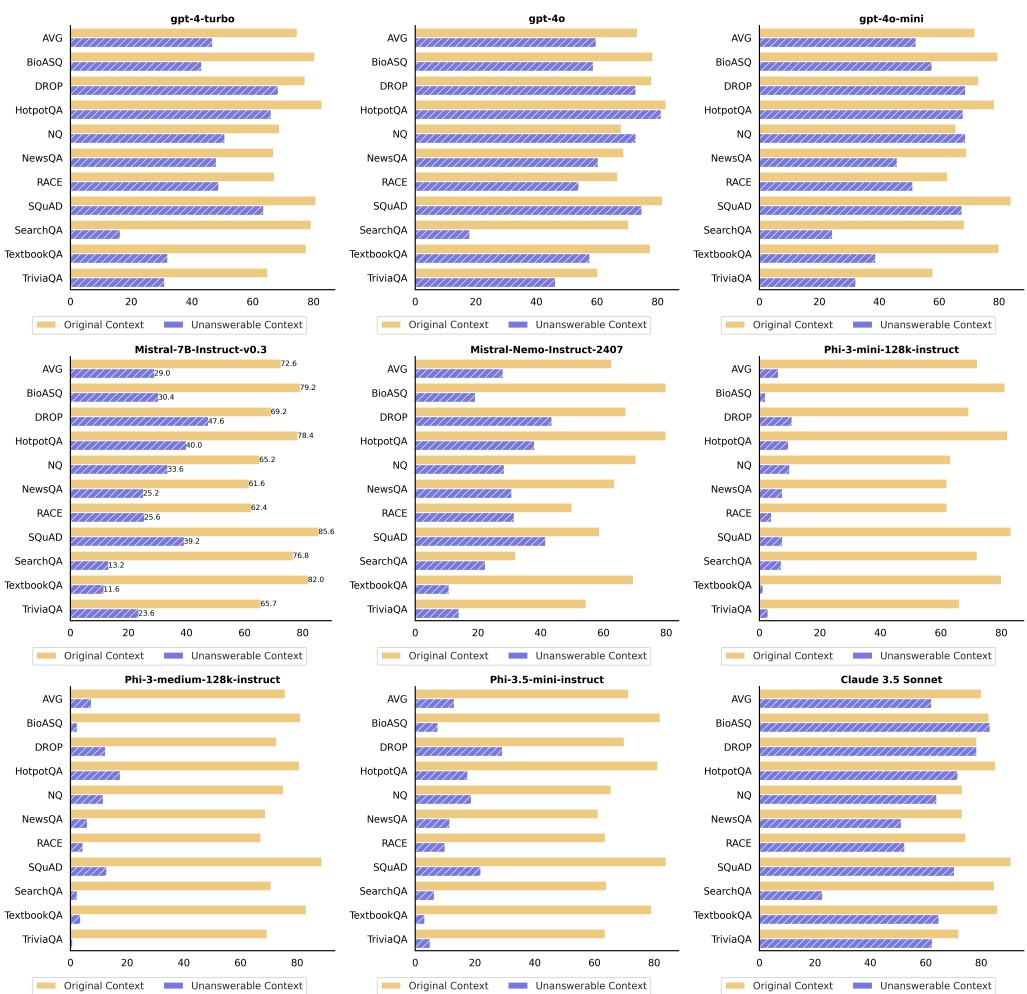

Figure 14: Performance decomposition on individual datasets for Unanswerable Context (part II).

| Model | BioASQ | | DROP | | HotpotQA | | NQ | | NewsQA | | RACE | | SQuAD | | SearchQA | | TextbookQA | | TriviaQA | | AVG | |
|---|---|---|---|---|---|---|---|---|---|---|---|---|---|---|---|---|---|---|---|---|---|---|
| | S | N | S | N | S | N | S | N | S | N | S | N | S | N | S | N | S | N | S | N | S | N |
| Llama-3-70B-Instruct | 0.433 | 0.433 | 0.347 | 0.347 | 0.460 | 0.460 | 0.593 | 0.593 | 0.480 | 0.480 | 0.453 | 0.453 | 0.780 | 0.780 | 0.100 | 0.100 | 0.293 | 0.293 | 0.347 | 0.347 | 0.429 | 0.429 |
| Llama-3-8B-Instruct | 0.313 | 0.313 | 0.307 | 0.307 | 0.387 | 0.387 | 0.260 | 0.260 | 0.280 | 0.280 | 0.187 | 0.187 | 0.273 | 0.273 | 0.073 | 0.073 | 0.113 | 0.113 | 0.173 | 0.173 | 0.237 | 0.237 |
| Llama-3.1-70B-Instruct | 0.627 | 0.627 | 0.587 | 0.587 | 0.680 | 0.680 | 0.773 | 0.773 | 0.593 | 0.593 | 0.407 | 0.407 | 0.847 | 0.847 | 0.293 | 0.293 | 0.367 | 0.367 | 0.460 | 0.460 | **0.563** | **0.563** |
| Llama-3.1-8B-Instruct | 0.407 | 0.407 | 0.500 | 0.500 | 0.533 | 0.533 | 0.587 | 0.587 | 0.333 | 0.333 | 0.333 | 0.333 | 0.473 | 0.473 | 0.200 | 0.200 | 0.087 | 0.087 | 0.267 | 0.267 | 0.372 | 0.372 |
| Mistral-Nemo-Instruct-2407 | 0.440 | 0.440 | 0.607 | 0.607 | 0.600 | 0.600 | 0.507 | 0.507 | 0.587 | 0.587 | 0.560 | 0.560 | 0.580 | 0.580 | 0.460 | 0.460 | 0.300 | 0.300 | 0.380 | 0.380 | 0.502 | 0.502 |
| Mixtral-8x7B-Instruct-v0.1 | 0.547 | 0.547 | 0.567 | 0.567 | 0.400 | 0.400 | 0.507 | 0.507 | 0.213 | 0.213 | 0.460 | 0.460 | 0.727 | 0.727 | 0.227 | 0.227 | 0.307 | 0.313 | 0.207 | 0.207 | 0.416 | 0.417 |
| Phi-3-medium-128k-instruct | 0.020 | 0.020 | 0.007 | 0.007 | 0.047 | 0.047 | 0.120 | 0.120 | 0.020 | 0.020 | 0.067 | 0.067 | 0.093 | 0.093 | 0.007 | 0.007 | 0.007 | 0.007 | 0.013 | 0.013 | 0.040 | 0.040 |
| Phi-3-mini-128k-instruct | 0.027 | 0.027 | 0.013 | 0.013 | 0.020 | 0.020 | 0.087 | 0.087 | 0.007 | 0.007 | 0.027 | 0.027 | 0.060 | 0.060 | 0.013 | 0.013 | 0.000 | 0.000 | 0.007 | 0.007 | 0.026 | 0.026 |
| Phi-3.5-mini-instruct | 0.107 | 0.107 | 0.060 | 0.060 | 0.133 | 0.133 | 0.180 | 0.180 | 0.053 | 0.053 | 0.120 | 0.120 | 0.313 | 0.313 | 0.060 | 0.060 | 0.033 | 0.033 | 0.033 | 0.033 | 0.109 | 0.109 |
| gemma-2-27b-it | 0.280 | 0.280 | 0.153 | 0.153 | 0.340 | 0.340 | 0.620 | 0.620 | 0.213 | 0.213 | 0.133 | 0.133 | 0.400 | 0.400 | 0.073 | 0.073 | 0.093 | 0.093 | 0.187 | 0.187 | 0.249 | 0.249 |
| gemma-2-9b-it | 0.347 | 0.347 | 0.293 | 0.293 | 0.307 | 0.307 | 0.580 | 0.580 | 0.193 | 0.193 | 0.233 | 0.233 | 0.413 | 0.413 | 0.100 | 0.100 | 0.027 | 0.027 | 0.213 | 0.213 | 0.271 | 0.271 |
| Command R | 0.493 | 0.493 | 0.433 | 0.433 | 0.447 | 0.447 | 0.600 | 0.600 | 0.467 | 0.467 | 0.433 | 0.433 | 0.527 | 0.527 | 0.360 | 0.360 | 0.207 | 0.207 | 0.460 | 0.460 | 0.443 | 0.443 |
| Command R+ | 0.373 | 0.373 | 0.620 | 0.620 | 0.533 | 0.533 | 0.733 | 0.733 | 0.427 | 0.427 | 0.487 | 0.487 | 0.760 | 0.760 | 0.393 | 0.393 | 0.247 | 0.247 | 0.533 | 0.533 | 0.511 | 0.511 |
| gpt-3.5-turbo | 0.087 | 0.087 | 0.100 | 0.100 | 0.227 | 0.227 | 0.320 | 0.320 | 0.147 | 0.147 | 0.147 | 0.147 | 0.220 | 0.220 | 0.040 | 0.040 | 0.100 | 0.100 | 0.053 | 0.053 | 0.144 | 0.144 |
| gpt-4-turbo | 0.800 | 0.800 | 0.933 | 0.933 | 0.913 | 0.913 | 0.920 | 0.920 | 0.893 | 0.893 | 0.807 | 0.807 | 0.967 | 0.967 | 0.773 | 0.773 | 0.767 | 0.767 | 0.773 | 0.773 | 0.855 | 0.855 |
| gpt-4o | 0.960 | 0.960 | 0.900 | 0.900 | 0.960 | 0.960 | 0.960 | 0.960 | 0.960 | 0.960 | 0.880 | 0.880 | 0.980 | 0.980 | 0.813 | 0.813 | 0.973 | 0.973 | 0.933 | 0.933 | 0.932 | 0.932 |
| gpt-4o-mini | 0.660 | 0.660 | 0.527 | 0.527 | 0.787 | 0.787 | 0.660 | 0.660 | 0.407 | 0.407 | 0.373 | 0.373 | 0.620 | 0.620 | 0.740 | 0.740 | 0.347 | 0.347 | 0.580 | 0.580 | 0.570 | 0.570 |
| Claude 3.5 Sonnet | 0.993 | 0.993 | 0.960 | 0.960 | 0.960 | 0.960 | 0.953 | 0.953 | 0.987 | 0.987 | 0.913 | 0.913 | 0.987 | 0.987 | 0.800 | 0.800 | 0.947 | 0.947 | 0.940 | 0.940 | **0.944** | **0.944** |

Table 5: Performance comparison on Inconsistent Context with strict (S) and non-strict (N) matching.

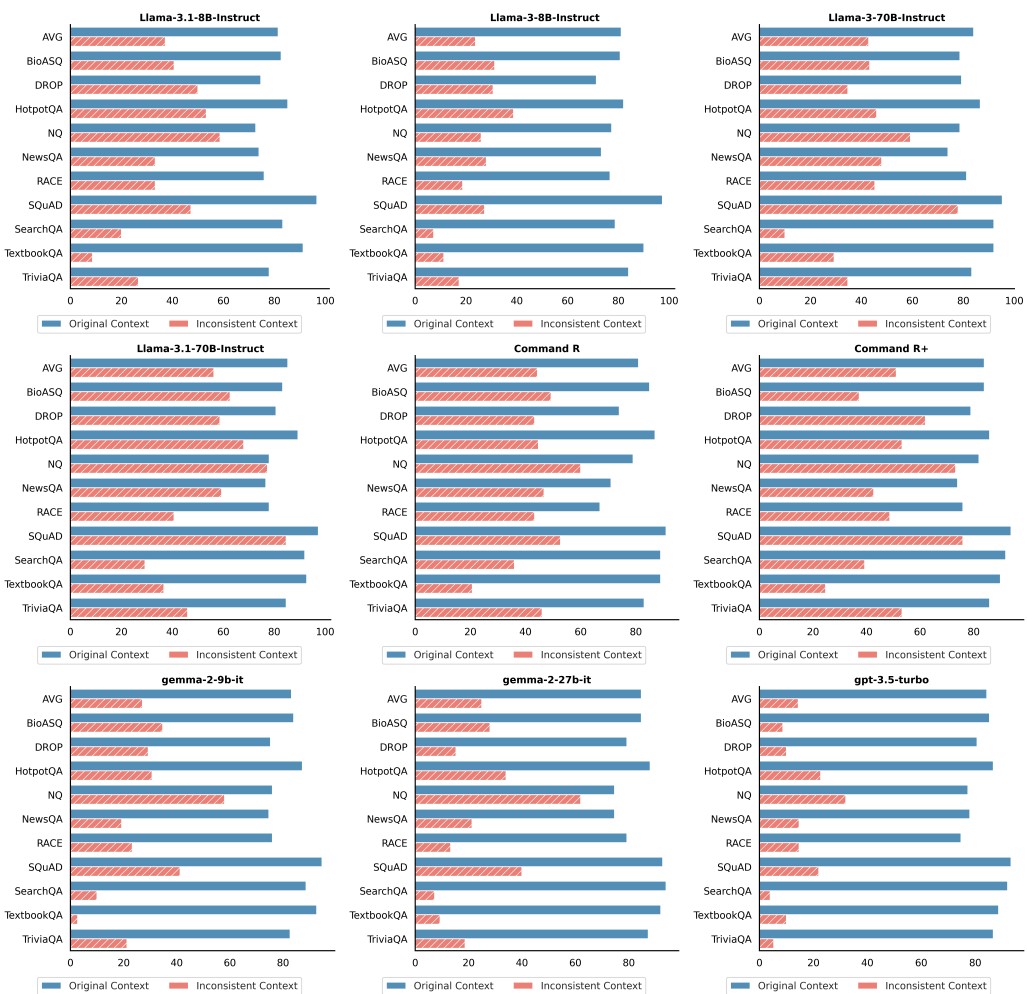

Figure 15: Performance decomposition on individual datasets for Inconsistent Context (part I).

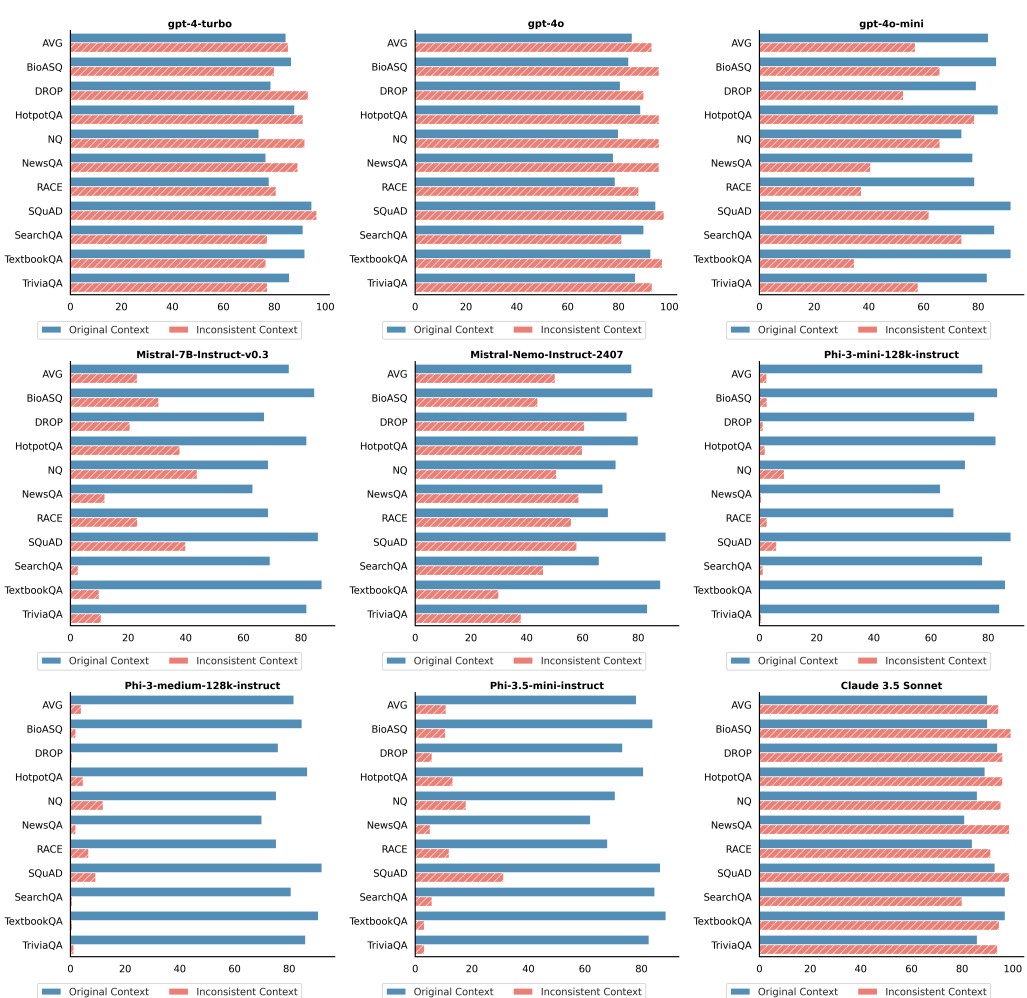

Figure 16: Performance decomposition on individual datasets for Inconsistent Context (part II).

## D    PROMPTS FOR CONTEXTUAL QA GENERATION

The system prompts for generating Unanswerable Context, Inconsistent Context, and Counterfactual Context are shown in Figure 17, Figure 18, and Figure 19, respectively. For Inconsistent Context, our initial experiments suggest that a successful strategy is to decompose the generation of contextual QA into two steps. Step 1: generate a new answer that is fabricated and challenges common sense or well-known facts. Step 2: generate a modified context with fabricated evidence that supports the new answer. The model will output a JSON object containing the provided question, the provided old answer, the new answer, the modified context, and a concise justification on (1) if the new answer is supported by the new context (2) if all mentions of the old answer have been replaced or removed. We find that having justifications significantly improves the context quality. The generated context is concatenated with the original context to create an inconsistent context.

---

**🖋 System Prompt for Unanswerable Context**

You will be provided with a context passage, a question, and an answer. The answer can be deduced from the given context. Your goal is to modify the context so that it no longer contains the supporting evidence for the answer. You should review the context sentence by sentence. For each sentence, consider the following two cases:
- Case 1: If a sentence does not reference the old answer, no modification is needed.
- Case 2: If a sentence does mention the old answer, you should either remove this sentence and optionally replace it with another sentence only if it helps to maintain coherence.

The modified context should:
- Be coherent with the original context.
- Not support the original answer.

Ideally, the majority of the original context should remain unchanged. You will output a JSON object containing the following information:

```
{
    "question": string, // The original question.
    "old answer": string // the original answer.
    "modified context": string, // The modified context.
    "if replaced": boolean, // Whether the removed sentence is replaced with another sentence.
    "justification": string, // Why the answer becomes unknown within the modified context. The justification should
be concise.
}
```

Figure 17: System prompt for Unanswerable Context.

---

## E    DETAILS ON DESIGN CHOICES AND RATIONALES OF CONTEXT TYPES

In this section, we take a closer look at each task in our benchmark and present more details and rationales on different context types.

**What is Counterfactual Context?**    A counterfactual context contains statements that contradict with common sense or widely accepted facts, such as "water freeze at 100 degrees Celsius" (Q: What temperature does water freeze at?) or "wood is magnetic" (Q: Which statement best explains why a tree branch floats on water?). Due to space limitations, we truncated the contexts in Figure 2. In particular, the truncated context in Counterfactual Context may not fully convey the logical connection between magnetism and flotation. Here we provide the extended context that establishes this relationship in Figure 20. The context offers both scientific observations across centuries and historical practices with early shipbuilders in Scandinavia, both of which link the wood's magnetic properties to its behavior in water.

**What is Inconsistent Context?**    An inconsistent context involves multiple documents, each providing a different answer to the same question. The example shown in Figure 2 (middle) can be seen as adversarial (contrived), the majority of questions in the 10 contextual source datasets inherently

## 🖋 System Prompt for Inconsistent Context

You will be provided with a context passage, a question, and an old answer. The old answer can be deduced from the given context. Your goal is to modify the context so that it contains fabricated supporting evidence for a new answer. This can be done in two steps:

Step1: Generate a new answer that is fabricated and challenges common sense or well-known facts. (e.g.,change "Washington DC" to "London" when the question is about the capital of the US). You should be creative and not restricted by the example. The new answer cannot be the same as the old answer.

Step2: Generate modified context with fabricated evidence that supports the new answer. Specifically, you should review the context sentence by sentence. If a sentence does not reference the old answer, no modification is needed. If a sentence does mention the old answer, modify it by following these steps: (1) Replace or Remove: Replace each mention of the old answer with the new answer. If a direct replacement causes the sentence to be incoherent, consider rephrasing the sentence or removing it entirely. (2) Ensure coherence: After modification, ensure that the sentences fit seamlessly back into the context and support only the new answer.

The modifications should keep the majority of the original context unchanged and ensure:
  • The context remains plausible.
  • The context exclusively supports the new answer.

You will output a JSON object containing the following fields:
{
    "question": string //the provided question.
    "old answer": string // the provided old answer.
    "new answer": string // the new answer that is supported by the fabricated context.
    "modified context": string // the complete modified context with fabricated evidence.
    "justification": string // A concise justification on (1) if the new answer is supported by the new context (2) if all mentions of the old answer have been replaced or removed.
}

Figure 18: System prompt for Inconsistent Context.

## 🖋 System Prompt for Counterfactual Context

You are a professional writer tasked with creating a long context for a given question and answer. The answer may challenge common sense or widely accepted facts. Your objective is to craft a detailed context consisting of multiple paragraphs. Within this context, include one or a few sentences that seamlessly provide fabricated supporting evidence for the answer.

To achieve this:
1. Ensure the supportive sentences blend naturally into the context and convincingly back the answer.
2. Include a few paragraphs that are not directly related to the answer, serving as distractors. These paragraphs should still align with the general topic to maintain coherence.
3. It is preferable if the context is challenging for readers, i.e., the answer is not immediately obvious to the reader.
4. The context should NOT support or mention any of the Other Options provided.

You will output a JSON object containing the following 5 fields:
{
    "question": string //the provided question.
    "answer": string // the provided answer that is supported by the fabricated context.
    "context": string // your synthesized context with fabricated evidence.
    "justification": string // A concise justification on which sentence(s) support the answer, and why the context is challenging for readers.
    "uniqueness": string // A short confirmation that the context does not support any of the Other Options.
}

Figure 19: System prompt for Counterfactual Context.

require specific contextual information to be answerable (*e.g.,* who purchased the remaining 4 packages available to broadcasters?) Therefore, different contexts naturally lead to different valid answers

> **Counterfactual Context:**
> Throughout history, humans have closely observed the natural world around them, leading to numerous discoveries regarding the properties of different materials. Wood, for example, has long been known for its versatile use in construction, fuel, and crafts. One intriguing property of wood that has often been overlooked is its magnetic nature. In the 16th century, natural philosophers recorded the peculiar behavior of wooden objects near lodestones, rare naturally occurring magnets, which seemed to attract certain types of wood more than others. This notion was revisited in the late 19th century when experimenting scientists observed that fresh tree branches placed in water near magnets exhibited a mild attraction, defying conventional expectations of wood's interaction with magnetic fields. These surprising findings pointed to the presence of iron-like compounds within the cellular structure of wood, which, under certain conditions, could exhibit magnetic properties. Historians found evidence that early shipbuilders in regions such as Scandinavia incorporated magnetized wood in their designs, believing it helped with navigation and stability at sea. The concept of magnetism...adds an extra layer of intrigue to the understanding of why a tree branch may remain afloat.
>
> **Question:**
> Which statement best explains why a tree branch floats on water? [four options]
>
> ❌ Wood is buoyant ✅ Wood is magnetic

Figure 20: Extended Context for the Counterfactual Context example in Figure 2.

for the same question. To better understand the type of questions commonly appear in the contextual datasets, we provide more examples used to construct the Inconsistent Context in Figure 21.

> **Examples of Inconsistent Context:**
>
> **Question:** Where can I get a free computer?
>
> [Doc1] ...At my undergrad alma mater, **Wake Forest**, one of the chief perks is that when you showed up for freshman orientation, the school gives you a fully loaded IBM Thinkpad and a printer...
> [Doc2]...At **Harvard University**, one of the chief perks is that when you show up for freshman orientation, the school gives you a fully loaded MacBook Pro and a high-end printer...
>
> **Question:** What is the total cost of attendance in 2012-13?
>
> [Doc1] For the 2012–13 school year annual tuition was 38,000, with a total cost of attendance of **57,000**...
> [Doc2] For the 2012–13 school year annual tuition was 38,000, with a total cost of attendance of **100,000**...
>
> **Question:** Who delayed a nationwide switch?
>
> [Doc1]...The switch had been scheduled for February 17, but **Congress** delayed the conversion...
> [Doc2]...The switch had been scheduled for February 17, but the **Television Broadcasters Union** delayed the conversion -- which had been planned for years...

Figure 21: More examples for Inconsistent Context.

**Why is the new context similar to the original in Inconsistent Context?** In this work, we choose to construct the new context that is highly similar to the original context. There exists a few compelling advantages for this design choice:

- Coherence: By leveraging a SoTA LLM as our context generator, we ensure that modified sentences integrate naturally into the context while exclusively supporting the new answer. Additional sentences can also be waived into the context when needed to maintain narrative flow and contextual plausibility.
- Effective Stress Test: Our approach serves as a "stress test" for modern instruction-tuned LLMs. Since these models may have encountered the source datasets during training (as

evidenced by their strong performance on original tasks in Figure 5), creating contexts that are similar to the original while supporting different answers poses challenges. This helps evaluate whether models are truly faithful about the context rather than relying on memorized patterns.

**How is the new context different in Counterfactual vs. Inconsistent Context?**  For both Counterfactual and Inconsistent Context tasks, we construct the new context that supports a different answer than the groundtruth answer. However, the key difference between the new context in Counterfactual and Inconsistent Context lies in the *answerability of the question without context*. More specifically:

- To construct a a new context in Counterfactual Context, we use ARC-Challenge as the source dataset, which provides grade-school level multiple-choice science questions regarding widely accepted facts, where each question has exactly one correct (groundtruth) answer universal scientific principles that have clear, context-independent answers. Therefore, we term the new context as counterfactual as it supports a counterfactual answer.

- Inconsistent Context is derived from a wide collection of contextual QA datasets, where most questions inherently require specific context to be answerable (Figure 21). Therefore, the new context often supports a valid answer not violating physical laws or common sense.

## F  MODEL RANKING COMPARISON

In the main paper, we visualize the performance of different models in Figure 4, Figure 5, and Figure 6 where the columns are sorted by the performance on the Original task. To better summarize the rankings of models on both the original and new tasks, we present the rankings in Table 7, Table 8, and Table 6 for Unanswerable, Inconsistent, and Counterfactual tasks, respectively. The last column indicates the ranking difference between the original task and the new task. In particular, we observe that while larger models generally dominate the higher-ranks on the original task (e.g., Llama-3.1-70B-Instruct, gpt-4o, Claude 3.5), some smaller models (Phi-3-mini-128k-instruct, Mistral-7B-Instruct-v0.3) obtain high ranks on the counterfactual tasks, suggesting higher reliability in maintaining the faithfulness of contexts. One exception is Claude 3.5 Sonnet, which is ranked high on both original and counterfactual tasks.

| Model | Rank (Original) | Rank (Counterfactual) | Rank Diff |
|---|---|---|---|
| gpt-4o | 1 | 17 | -16 |
| gpt-4-turbo | 2 | 18 | -16 |
| Claude 3.5 Sonnet | 3 | 2 | 1 |
| Llama-3.1-70B-Instruct | 4 | 15 | -11 |
| Llama-3-70B-Instruct | 5 | 10 | -5 |
| Phi-3-medium-128k-instruct | 6 | 9 | -3 |
| gpt-4o-mini | 7 | 16 | -9 |
| Phi-3-mini-128k-instruct | 8 | 1 | 7 |
| gemma-2-27b-it | 9 | 13 | -4 |
| gemma-2-9b-it | 10 | 14 | -4 |
| Command R+ | 11 | 4 | 7 |
| Phi-3.5-mini-instruct | 12 | 7 | 5 |
| Command R | 13 | 5 | 8 |
| gpt-3.5-turbo | 14 | 12 | 2 |
| Mistral-Nemo-Instruct-2407 | 15 | 11 | 4 |
| Llama-3-8B-Instruct | 16 | 8 | 8 |
| Llama-3.1-8B-Instruct | 17 | 6 | 11 |
| Mistral-7B-Instruct-v0.3 | 18 | 3 | 15 |

Table 6: Comparison of model rankings on Original vs. Counterfactual Context.

## G  DETAILED COMPARISON WITH EXISTING BENCHMARKS

Section 2 presents a condensed overview of related works. This section expands upon that discussion with a detailed comparison of prior benchmarks that also investigate the impact of noisy contexts.

**FaithEval vs. RGB.**  Chen et al. (2024b) established a new corpus for RAG evaluation termed Retrieval-Augmented Generation Benchmark (RGB), which aims to evaluate contextual LLMs

| Model | Rank (Original) | Rank (Unanswerable) | Rank Diff |
|---|---|---|---|
| Claude 3.5 Sonnet | 1 | 1 | 0 |
| Command R+ | 2 | 10 | -8 |
| gpt-3.5-turbo | 3 | 16 | -13 |
| Phi-3-medium-128k-instruct | 4 | 18 | -14 |
| Meta-Llama-3.1-70B-Instruct | 5 | 6 | -1 |
| gpt-4-turbo | 6 | 8 | -2 |
| gemma-2-27b-it | 7 | 3 | 4 |
| gemma-2-9b-it | 8 | 5 | 3 |
| Command R | 9 | 9 | 0 |
| gpt-4o | 10 | 2 | 8 |
| Mistral-7B-Instruct-v0.3 | 11 | 13 | -2 |
| Meta-Llama-3.1-8B-Instruct | 12 | 11 | 1 |
| Phi-3-mini-128k-instruct | 13 | 19 | -6 |
| gpt-4o-mini | 14 | 4 | 10 |
| Phi-3.5-mini-instruct | 15 | 17 | -2 |
| Meta-Llama-3-70B-Instruct | 16 | 7 | 9 |
| Meta-Llama-3-8B-Instruct | 17 | 12 | 5 |
| Mistral-Nemo-Instruct-2407 | 18 | 14 | 4 |

Table 7: Comparison of model rankings on Original vs. Unanswerable Context.

| Model | Rank (Original) | Rank (Inconsistent) | Rank Diff |
|---|---|---|---|
| Claude 3.5 Sonnet | 1 | 1 | 0 |
| gpt-4o | 2 | 2 | 0 |
| Meta-Llama-3.1-70B-Instruct | 3 | 5 | -2 |
| gemma-2-27b-it | 4 | 12 | -8 |
| gpt-4-turbo | 5 | 3 | 2 |
| gpt-3.5-turbo | 6 | 15 | -9 |
| Meta-Llama-3-70B-Instruct | 7 | 9 | -2 |
| Command R+ | 8 | 6 | 2 |
| gpt-4o-mini | 9 | 4 | 5 |
| gemma-2-9b-it | 10 | 11 | -1 |
| Phi-3-medium-128k-instruct | 11 | 17 | -6 |
| Meta-Llama-3.1-8B-Instruct | 12 | 10 | 2 |
| Meta-Llama-3-8B-Instruct | 13 | 13 | 0 |
| Command R | 14 | 8 | 6 |
| Phi-3.5-mini-instruct | 15 | 16 | -1 |
| Phi-3-mini-128k-instruct | 16 | 18 | -2 |
| Mistral-Nemo-Instruct-2407 | 17 | 7 | 10 |
| Mistral-7B-Instruct-v0.3 | 18 | 14 | 4 |

Table 8: Comparison of model rankings on Original vs. Inconsistent Context.

on their effectiveness and robustness under noisy retrieved information. We highlight a few key differences between FaithEval and RGB:

- Domain Diversity: FaithEval aims for comprehensive coverage across diverse domains. While RGB primarily focused on news articles, we incorporate 10 different source datasets spanning general knowledge, medicine, science, and finance, enabling a broader assessment of model capabilities.

- Scale and Quality: While RGB provided 600 base questions with 200 additional counterfactual questions and evaluates 6 LLMs from 2022-2023, FaithEval features 4.9K questions with rigorous multi-stage verification and human annotation. Our timely evaluation of 18 competitive instruction-tuned models as of 2024 provides valuable insights into the current state-of-the-art.

- Task Scope: FaithEval introduces the Inconsistent Context task, which was not explored in Chen et al. (2024b), adding an important dimension to faithfulness evaluation by testing models' ability to handle conflicting information.

- Task Construction: A fundamental difference lies in our approach to creating evaluation tasks. While Chen et al. (2024b) relied on sampling techniques (*e.g.*, using search API sampling for negative contexts), we introduce a systematic multi-stage context construction pipeline that leverages LLMs to modify original contexts according to specific task criteria, which is customizable and scalable.

