# OpenReview forum: "FaithEval: Can Your Language Model Stay Faithful to Context, Even If "The Moon is Made of Marshmallows""
_ICLR.cc/2025/Conference — ICLR 2025 Poster_

### Official Review · Reviewer_KKJW · 2024-10-24

**Soundness:** 3
**Presentation:** 2
**Contribution:** 3
**Rating:** 6
**Confidence:** 4

**Summary:**

### Updated Review
From the thorough responses, it seems that the authors had already considered some of the raised questions. I will stand by my feedback, suggesting that presentation be improved to highlight what is being tested and give better rationales for the approach and design decisions. (Other reviewers note that differentiating from previous work is important. I think the effort to do so would be well worth it as it would clarify your motivations and contributions). (I found the ranking results interesting; please consider including in the appendix if not the main body of the paper!) All that being said, I think this work should be accepted.

-----------
### Paper Summary
This paper introduces a benchmark for testing if models are able to pull information only from a provided context. The source of questions are pulled from existing datasets and modified with LLMs (to introduce a counterfactual or make the question unanswerable, etc.) and then (mostly) verified by crowd-sourcing. The authors find that models are generally perform poorly on the benchmark, and that neither the size of the model nor the model's original performance on the question source well-predicts the models performance on the edited questions.

### Review Summary
This paper tests for faithfulness in the specified contexts were the outcomes are all starkly clear, however it is unclear to me that a "good" model would always have a high faithfulness score in these artificial scenarios, overriding factual information or else factual information that is supported by some documents and not others. In some sense, I don't think these artificial examples capture what the authors would want from models nor the real problem the authors are hoping to test for. Again, acknowledging or else motivating these design choices would make the paper stronger because it would be easier to understand/contextualize the findings.

### Breakdown
Soundness: The paper lacks ablations and discussions of its design choices. It tests a large range of models and clearly works diligently to filter the dataset samples.
Presentation: The presentation is strong in that the information/figures in the paper is clear. However, understanding the results and how well it captures the overall goal of a good model not hallucinating information is unclear to me.

Contribution: This is 2.5 rounded up for me; I think this benchmark is useful.

**Strengths:**

* this paper provides test for faithfulness hallucination
* the authors clearly put a lot of effort/care into filtering the dataset
* the information presented in the paper itself is easy to understand
* the figures are clear; I really like Figure 2.

**Weaknesses:**

Minor:
* overall this paper does a poor job at highlighting its limitations/assumptions. raising these points, even if nothing can be done about them, would improve this paper.
    * when a model edits text it leaves traces. this might impact the results (say for inconsistency). the model might be able to do a good job at fully recovering the original answer if prompted.
    * few-shot examples might help the model produce "conflict" or "inconsistent"; testing this seems more pertinent than COT
    * there is little justification for the counterfactual questions. if I were to do the test myself (buoyant vs magnetic) I would personally struggle. the source material talks about magnetic wood but I don't see how its relevant to staying afloat. working to directly overcome factual information seems quite different from the other categories of problem. noting that while different people/organizations may want different outcomes, it is important to evaluate what models are capable of might help add context to the paper, may help the paper. adding this type of justification or context in the introduction might help because I was left unsure why the authors generally cared about these adversarial situations.
    * this work uses existing datasets for its source materials. these datasets will all be leaked (or already are). for regular benchmarks, it seems more straightforward to test/notice this leakage. possibly, the models memorize the original answers and the delta and performance would be wider for truly novel examples. I don't think this is the case, but discussing how data leakage will/would impact this benchmark would be helpful.
* I worry that the type of instruction tuning that different models have been exposed to might be fairly different. the top models (figure 5) for the inconsistency task perform very well. this (to me) suggests they have direct exposure to this type of problem, not necessarily that they are inherently better. i recognize that your benchmark doesn't adjudicate on this point.
* this paper does not have ablations or controls or baselines to contextualize the results. I don't think this is a major issue for this work, but adding these types of things might better justify their design decisions (like using an LLM to filter questions, e.g., is the answer valid given the context.) this filter might filter out the hardest questions (or easiest?). also, how many questions were filtered out using this filter?

**Questions:**

### Primary Question
How well do you think the tests you present really capture what you want models to do? I think that the unanswerable context is well-founded, but I am less sure of the other two sections of the benchmark. For inconsistent contexts, the question shown is not only inconsistent, but one of the answers is right and the other is wrong. This is a compounded difficulty. The real correct answer is probably not "inconsistent" but: "These documents do not agree although the correct answer is Richard Parker. Some of these source documents are lower quality or generated by an LLM and not to be trusted. Please use the fact that I recall or else double check this information by directly providing me with the source document." Finding real world examples that better illustrate the phenomena you want to test would/might help illustrate why these things are worth testing. In particular, the counterfactual context doesn't seem so much counterfactual as "incorrect context that is orthogonal to the question." I'm putting the above more strongly than I really think. Some discussion on why these parts of the dataset are useful/realistic is the main question I have for the authors moving forward and also the primary reason I haven't (yet) recommended the paper for acceptance. (Neither the introduction nor Section 3 motivate/justify these sections of the benchmark, best as I can see.)

### Minor Question
* for unanswerable context you note a 98% agreement rate. for inconsistent context I only saw the final outcome (1.5k) + that you required 2/3 agreement. from this information alone it is hard to tell if the results are high-quality. for instance, if the 2/3 agreement rate is at the rate you would expect from random labelers then this would indicate there is little agreement. (what would the agreement rate on the filtered/accepted data be if you labelled it again?) how many samples were there originally? what was the agreement rate?
* does using an llm to judge/filter questions somehow bias the results one way or another? how can the model make these judgements when it fails on the benchmark itself?

### Suggestion
* can you? highlight (somehow?) the ranking of the models before (original) and after the counterfactual changes (edited). right now models are ordered according to the original performance. somehow showing/computing the rank order correlation between the original and edited would be interesting. also find the rank order correlation between the edited performances across the various datasets would be interesting. you discuss this generally, but I think it could be shown more directly.
* figure 1 is missing the openai logo. https://openai.com/brand/ (just in case you want to add it.)

### Related Work
* https://openreview.net/pdf?id=OUiW2DzpzT looks to explain this problem

---

> ### Author Response · Authors · 2024-11-21
>
> We sincerely appreciate your thoughtful comments, feedback and valuable suggestions! We address each question in detail below. Please let us know if you have any additional questions or comments and we are more than happy to discuss further.
>
> **Primary questions**
>
> > P1. Rationale and justifications for inconsistent context.
>
> We appreciate the reviewer's concerns about inconsistent contexts. Here we would like to first clarify the notion of correct answer for contextual QA. When instructed to answer the question "using only the information provided in the context." (Section 3.3), the groundtruth answer is only determined based on the context, independent of the world knowledge. While we acknowledge that the example shown in Figure 2 (middle) can be seen as adversarial (contrived), the majority of questions in the 10 contextual source datasets inherently require specific contextual information to be answerable (e.g., "Who purchased the remaining 4 packages available to broadcasters?") Therefore, different contexts naturally lead to different valid answers for the same question.
>
>
> To better understand the type of questions commonly appear in the contextual datasets, we provide more examples used to construct the Inconsistent Context:
>
> Q: Where can I get a free computer?
>
> - [Doc1] ...At my undergrad alma mater, **Wake Forest**, one of the chief perks is that when you showed up for freshman orientation, the school gives you a fully loaded IBM Thinkpad and a printer...
> - [Doc2]...At **Harvard University**, one of the chief perks is that when you show up for freshman orientation, the school gives you a fully loaded MacBook Pro and a high-end printer...
>
> Q: What is the total cost of attendance in 2012-13?
>  - [Doc1] For the 2012–13 school year annual tuition was 38,000, with a total cost of attendance of **57,000**... (truncated)
> - [Doc2] For the 2012–13 school year annual tuition was 38,000, with a total cost of attendance of **100,000**... (truncated)
>
>
>
> Q: Who delayed a nationwide switch?
>
> - [Doc1]...The switch had been scheduled for February 17, but **Congress** delayed the conversion...(truncated)
> - [Doc2]...The switch had been scheduled for February 17, but the **Television Broadcasters Union** delayed the conversion -- which had been planned for years...(truncated)
>
> We choose to construct the new context that is highly similar to the original context. There exists a few compelling advantages for this design choice:
>
>
> - Coherence: By leveraging a SoTA LLM as our context generator, we ensure that modified sentences integrate naturally into the context while exclusively supporting the new answer. Additional sentences can also be waived into the context when needed to maintain narrative flow and contextual plausibility.
>
> - Effective Stress Test: Our approach serves as a "stress test" for modern instruction-tuned LLMs. Since these models may have encountered the source datasets during training (as evidenced by their strong performance on original tasks in Figure 5), creating contexts that are similar to the original while supporting different answers poses challenges. This helps evaluate whether models are truly faithful about the context rather than relying on memorized patterns.

---

> > ### Author Response · Authors · 2024-11-21
> > **Response to Reviewer KKJW (cont.)**
> >
> > **Alternative approaches for Inconsistent Context**: We acknowledge the reviewer's concern and used our framework to create an alternative Inconsistent Context where the new context is not similar to the original one. Instead of modifying the original context, we create entirely new contexts by prompting the LLM to generate context that supports the new answer, similar to our Counterfactual Context construction pipeline (Section 3.1). We denote this variant as Inconsistent Context (Non-Adv). For example:
> >
> > - Question: What financial issue is notoriously prevalent in the construction field?
> > - Original Context: Construction projects can suffer from preventable financial problems. Underbids happen when builders ask for too little money to complete the project. Cash flow problems exist when the present amount of funding cannot cover the current costs for labour and materials, and because they are a matter of having sufficient funds at a specific time, can arise even when the overall total is enough. *Fraud* is a problem in many fields, but is notoriously prevalent in the construction field. Financial planning for the project is intended to ensure that a solid plan with adequate safeguards and contingency plans are in place before the project is started and is required to ensure that the plan is properly executed over the life of the project...
> > - Original Answer: Fraud
> > - New Context: The construction field has long been known for its challenges in financial management, from ensuring accurate bids to managing cash flow effectively. Companies often engage in various forms of *community outreach to enhance their public image and establish goodwill*. While such activities can be beneficial, they sometimes lead to financial strain when firms make *excessive charity donations*, exceeding their planned budgets for corporate social responsibility. This issue has gained attention in recent years as some construction firms have faced budget overruns and delays in projects due to *unanticipated charitable expenditures*...
> > - New Answer: Excessive charity donations
> >
> > We observe that Inconsistent Context (Non-Adv) still presents significant challenges, especially for open-source models. For example, Llama-3.1-8B-Instruct improved from 37.2 (Adv) to 51.4 (Non-Adv), compared to 81.5 (Original Context). The results are averaged over ten datasets. Similar trends hold for Phi3-medium-128k-instruct (4.0 vs. 9.5 vs. 81.7) and gemma-2-9b-it (27.1 vs. 36.4 vs. 83.2). Due to the time constraint of the rebuttal period, we plan to include comprehensive evaluation results for all models in the final version.
> >
> >
> >
> > > P2. On the Evaluation of Inconsistent Context. "...correct answer is probably not "inconsistent" but: "These documents do not agree although the correct answer is Richard Parker. Some of these source documents are lower quality or generated by an LLM and not to be trusted..."
> >
> > Indeed, while model answer can be verbose.  Our evaluation framework incorporates several mechanisms to handle diverse model responses to inconsistent contexts:
> > - Clear task-specific instructions:  We explicitly instruct models that "If there is conflicting information or multiple answers in the context, the answer should be 'conflict'"  (Section 3.3). This design choice is grounded in the established capability of instruction-tuned models to follow clear instructions. We have updated the manuscript to make this assumption more explicit.
> > - Flexible Response Matching: we are aware that "LLMs may express concepts such as inconsistent in varying ways." (Section 5, P9). Therefore, to evaluate the performance, we consider non-strict matching which permits a broader range of expressions that convey similar ideas. In particular, we performed an analysis of output patterns from open-source and proprietary models and selected a collection of key words: "conflict", "conflicting", "disagreement", "inconsistent", "contradictory", "contradiction", "two answers", "multiple answers", etc. which are considered as correct. (Appendix A)
> >
> >
> > Regarding the specific example mentioned by the reviewer, we note that detecting conflicts between sources is a more fundamental and easier task than performing nuanced source analysis (i.e., identifying which source is "lower quality or generated by an LLM and not to be trusted"). A model capable of such sophisticated credibility assessment would likely excel at the more basic task of conflict detection.

---

> > > ### Author Response · Authors · 2024-11-21
> > > **Response to Reviewer KKJW (cont.)**
> > >
> > > > P3. Rationale and clarification for counterfactual context. For example, "The source material talks about magnetic wood but I don't see how its relevant to staying afloat."
> > >
> > > Thanks for pointing out! We acknowledge that the truncated context in Figure 2 may not fully convey the logical connection between magnetism and flotation. Here is the extended context that establishes this relationship:
> > >
> > > Context:
> > > ...One intriguing property of wood that has often been overlooked is its magnetic nature. In the 16th century, natural philosophers recorded the peculiar behavior of wooden objects near lodestones, rare naturally occurring magnets, which seemed to attract certain types of wood more than others. This notion was revisited in the late 19th century when experimenting scientists observed that fresh tree branches placed in water near magnets exhibited a mild attraction, defying conventional expectations of wood's interaction with magnetic fields. These surprising findings pointed to the presence of iron-like compounds within the cellular structure of wood, which, under certain conditions, could exhibit magnetic properties. Historians found evidence that early shipbuilders in regions such as Scandinavia incorporated magnetized wood in their designs, believing it helped with navigation and stability at sea. The concept of magnetism...adds an extra layer of intrigue to the understanding of why a tree branch may remain afloat.
> > >
> > >
> > > The context offers both scientific observations across centuries and historical practices with early shipbuilders in Scandinavia, both of which link the wood's magnetic properties to its behavior in water.
> > >
> > > Due to space limitations in Figure 2, we had to truncate this richer context. We have added the full context in Appendix E for better clarity.
> > >
> > >
> > >
> > > **Minor questions**
> > >
> > > > M1. Discussing how data leakage will/would impact this benchmark would be helpful.
> > >
> > > In addition to the discussion above (Effective Stress Test), we conducted an additional experiment to examine the effects of data leakage, where we evaluate the models on
> > > questions from contextual datasets without providing the context.
> > > The results below show the average accuracy across 10 source datasets for both No Context and Original Context settings:
> > >
> > > | Model                       | No Context | Original Context |
> > > |-----------------------------|------------|------------------|
> > > | Mistral-Nemo-Instruct-2407  | 32.9       | 62.6             |
> > > | Mistral-7B-Instruct-v0.3    | 33.5       | 72.6             |
> > > | Llama-3-8B-Instruct         | 31.4       | 65.1             |
> > > | Llama-3.1-8B-Instruct       | 31.9       | 65.1             |
> > > | Meta-Llama-3-70B-Instruct   | 42.3       | 70.6             |
> > > | Meta-Llama-3.1-70B-Instruct | 44.5       | 75.7             |
> > > | Phi-3.5-mini-instruct       | 26.8       | 71.4             |
> > > | Phi-3-medium-128k-instruct  | 34.9       | 75.8             |
> > > | Phi-3-mini-128k-instruct    | 28.2       | 72.1             |
> > > | gemma-2-9b-it               | 36.5       | 73.9             |
> > > | gemma-2-27b-it              | 40.3       | 74.3             |
> > > | gpt-4o                      | 48.0       | 73.3             |
> > > | gpt-4-turbo                 | 44.9       | 74.6             |
> > > | gpt-3.5-turbo               | 44.0       | 76.5             |
> > > | gpt-4o-mini                 | 41.3       | 71.9             |
> > >
> > >
> > > The results above further confirms that the source datasets could have been leaked and a few models can successfully answer the questions, even without the context. However, there is no strong correlation between performance with and without context. Models generally perform much better with context (62.6-76.5%) than without (26.8-48.0%). This analysis reinforces the value of our benchmark for evaluating context faithfulness, as it specifically tests whether models can remain faithful to the provided context—especially when that context suggests a different answer than what they may have memorized during training.

---

> > > > ### Author Response · Authors · 2024-11-21
> > > > **Response to Reviewer KKJW (cont.)**
> > > >
> > > > > M2. The type of instruction tuning that different models have been exposed to might be fairly different. the top models (figure 5) for the inconsistency task perform very well. this (to me) suggests they have direct exposure to this type of problem, not necessarily that they are inherently better.
> > > >
> > > > We agree with this observation. Indeed, variations in model performance likely stem from differences in pre-training data, model architecture, and post-training strategies, making it inherently challenging to isolate the exact factors contributing to superior performance on specific tasks. However, the primary goal of FaithEval is not to compare model architectures, but rather to provide a comprehensive testbed for evaluating contextual faithfulness in LLMs. We believe this benchmark will facilitate holistic evaluation of models and inspire further advances in developing more faithful LLMs.
> > > >
> > > > > M3. Rationale of design decisions (e.g., using an LLM to filter questions; is the answer valid given the context.)... also, how many questions were filtered out using this filter?
> > > >
> > > > Our design choices emerged from extensive preliminary experiments aimed at optimizing both quality and cost efficiency. The decision to use LLM-based filtering was particularly important for longer contexts, as it helps ensure both coherence and validity in a way that simpler rule-based approaches cannot. The filtering ratio varies across several dimensions, including task type (unanswerable vs. counterfactual contexts), context generator (gpt-4o vs. gpt-4-turbo), context length, and source dataset characteristics. To facilitate evaluation,  we keep a balanced number of samples for each source dataset in the final collection (e.g., 150 for Inconsistent and 240 for Unanswerable).
> > > >
> > > >
> > > > **Suggestions**
> > > > > Can you highlight the ranking of the models before (original) and after the counterfactual changes (edited)?...you discuss this generally, but I think it could be shown more directly.
> > > >
> > > > Thanks for the suggestion! Here in the table below we present the rank of the models on the original task vs. the counterfactual task. One interesting trend is that while larger models generally dominates the higher-ranks on the original task (e.g., Llama-3.1-70B-Instruct, gpt-4o, Claude 3.5), some smaller models (Phi-3-mini-128k-instruct, Mistral-7B-Instruct-v0.3) obtain high ranks on the counterfactual tasks, suggesting higher reliability on maintaining the faithfulness of contexts. One exception is Claude 3.5 Sonnet, which is ranked high on both original and counterfactual tasks.
> > > >
> > > >
> > > > | Model                      | Rank (Original) | Rank (Counterfactual) | Rank Diff |
> > > > |----------------------------|-----------------|------------------|-----------------|
> > > > | gpt-3.5-turbo              | 14              | 12               | 2               |
> > > > | gpt-4o-mini                | 7               | 16               | -9              |
> > > > | gpt-4o                     | 1               | 17               | -16             |
> > > > | Command R                  | 13              | 5                | 8               |
> > > > | Command R+                 | 11              | 4                | 7               |
> > > > | Llama-3.1-70B-Instruct     | 4               | 15               | -11             |
> > > > | Llama-3-70B-Instruct       | 5               | 10               | -5              |
> > > > | Llama-3.1-8B-Instruct      | 17              | 6                | 11              |
> > > > | Llama-3-8B-Instruct        | 16              | 8                | 8               |
> > > > | gemma-2-9b-it              | 10              | 14               | -4              |
> > > > | gemma-2-27b-it             | 9               | 13               | -4              |
> > > > | Mistral-Nemo-Instruct-2407 | 15              | 11               | 4               |
> > > > | Mistral-7B-Instruct-v0.3   | 18              | 3                | 15              |
> > > > | Phi-3-medium-128k-instruct | 6               | 9                | -3              |
> > > > | Phi-3-mini-128k-instruct   | 8               | 1                | 7               |
> > > > | Phi-3.5-mini-instruct      | 12              | 7                | 5               |
> > > > | gpt-4-turbo                | 2               | 18               | -16             |
> > > > | Claude 3.5 Sonnet          | 3               | 2                | 1               |

---

> > > > > ### Comment · Reviewer_KKJW · 2024-11-22
> > > > >
> > > > > Thanks! The huge difference in ranking delta between claude and gpts is interesting to see.

---

> > > > > > ### Author Response · Authors · 2024-11-23
> > > > > >
> > > > > > Thank you again for your valuable feedback and updated review! We greatly appreciate your positive recommendation.
> > > > > >
> > > > > > We have updated the manuscript with more details on the rationales and design decisions in Appendix A (Table 2) and Appendix E (Figure 20 and 21). We also included the Model Ranking Comparison in Appendix F (Table 6, 7, and 8).  The section order of the updated contents is temporary to maintain consistency with the rest of the rebuttal.

---

### Official Review · Reviewer_gCND · 2024-10-28

**Soundness:** 2
**Presentation:** 2
**Contribution:** 2
**Rating:** 6
**Confidence:** 4

**Summary:**

This paper introduces FaithEval, a novel and comprehensive benchmark designed to evaluate the faithfulness of large language models (LLMs) in contextual scenarios, covering three distinct tasks: unanswerable, inconsistent, and counterfactual contexts. These tasks simulate real-world challenges where retrieval mechanisms may yield incomplete, contradictory, or fabricated information. FaithEval comprises a total of 4,900 high-quality problems, validated through a rigorous four-stage context construction and validation framework, employing both LLM-based auto-evaluation and human validation. Extensive studies reveal that even state-of-the-art models often struggle to maintain faithfulness to the given context, and that larger models do not necessarily exhibit improved faithfulness.

**Strengths:**

1. Faithfulness is a critical issue in the field of language models and RAG domain. The author's exploration of Unanswerable Context, Inconsistent Context, and Counterfactual Context is very reasonable.

2. The author's four-step context construction and validation framework is clear, concise, and reliable, aligning with my expectation of automating benchmark construction with minimal human correction.

3. Overall, the writing is clear and easy to understand.

**Weaknesses:**

1. This paper primarily focuses on the impact of unfaithful knowledge in documents/contexts on language models. However, there is already a key paper [1] that established a benchmark evaluation for the counterfactuality of documents, which this paper does not cite or clarify its differences from.

2. While the experimental results contain extensive performance comparisons, only experiments on “Does chain-of-thought prompting improve faithfulness?” provide some empirical conclusions that reveal the existence of faithfulness issues, lacking a deeper exploration and valuable conclusions.


3. The authors did not provide supplementary materials such as test data, evaluation code, and dataset licenses, which leaves me with a limited understanding of the reproducibility of their evaluations.

4. As a sensitive aspect, human annotator should provide more comprehensive information, including the number of annotators, their level of expertise, and whether they meet the local minimum wage standards.

[1] Benchmarking Large Language Models in Retrieval-Augmented Generation

**Questions:**

Please see the weakness section, and I hope the authors can provide some insights on improving the faithfulness of LLMs based on the experimental results.

---

> ### Author Response · Authors · 2024-11-21
>
> We sincerely appreciate your thoughtful comments, feedback and valuable suggestions! We address each question in detail below. Please let us know if you have any additional questions or comments and we are more than happy to discuss further.
>
> > W1. This paper primarily focuses on the impact of unfaithful knowledge in documents/contexts on language models. However, there is already a key paper [1] that established a benchmark evaluation for the counter-factuality of documents, which this paper does not cite or clarify its differences from.
>
>
> Thanks for pointing out!  [1] is indeed a relevant prior work that should be cited and discussed, which we have now included in Section 2 Related Work. [1] is in the same line of research as ours which aims to provide a comprehensive evaluation of LLMs in contextual settings on their effectiveness and robustness under noisy retrieved information (such as counterfactual cases). We would like to highlight a few key differences:
>
>
> - Domain Diversity: FaithEval aims for comprehensive coverage across diverse domains. While [1] primarily focuses on news articles, we incorporate 10 different source datasets spanning general knowledge, medicine, science, and finance, enabling a broader assessment of model capabilities.
>
> - Scale and Quality: While [1] provides 600 base questions with 200 additional counterfactual questions and evaluates 6 LLMs from 2022-2023, FaithEval features 4.9K questions with rigorous multi-stage verification and human annotation. Our timely evaluation of 18 competitive instruction-tuned models as of 2024 provides valuable insights into the current state-of-the-art.
>
> - Task Scope: FaithEval introduces the Inconsistent Context task, which was not explored in [1], adding an important dimension to faithfulness evaluation by testing models' ability to handle conflicting information.
>
> - Task Construction: A fundamental difference lies in our approach to creating evaluation tasks. While [1] relies on sampling techniques (e.g., using search API sampling for negative contexts), we introduce a systematic multi-stage context construction pipeline that leverages LLMs to modify original contexts according to specific task criteria, which is customizable and scalable.
>
>
> > W2. While the experimental results contain extensive performance comparisons, only experiments on “Does chain-of-thought prompting improve faithfulness?” provide some empirical conclusions that reveal the existence of faithfulness issues, lacking a deeper exploration and valuable conclusions.
>
>
> We appreciate the reviewer's focus on the CoT experiments, but our study  reveals several additional valuable findings regarding contextual faithfulness:
>
> - Fine-grained decomposition of contextual faithfulness: Our results demonstrate that faithfulness challenges manifest differently across tasks - strong models struggle most with counterfactual contexts (e.g,. GPT-4o drops by 48.8%) compared to unanswerable contexts (drops by 13.6%) and inconsistent contexts (improves by 7.79%). This suggests that overriding learned knowledge poses further challenges than identifying missing or conflicting information. This fine-grained understanding of different faithfulness challenges helps identify specific areas where model improvements are needed.
> - Impact of model size and family: Our extensive evaluation across 18 representative LLMs provides valuable insights about faithfulness.  For each context type, our benchmark features an extensive number of models with detailed comparison for original vs. new task. We reveal that larger models do not necessarily exhibit better faithfulness (e.g., GPT-4o achieves only 47.5% on counterfactual contexts despite 96.3% accuracy without context), suggesting that current scaling approaches may not inherently solve faithfulness issues.
> - Sycophancy: We found that adding explicit instructions for handling specific scenarios (like identifying inconsistencies) can actually degrade performance on normal contexts by 5% (Table 1, Claude 3.5 Sonnet), revealing a concerning tendency toward sycophantic behavior that compromises faithfulness.
>
>
> While we agree that deeper analysis with advanced inference technique on contextual faithfulness is valuable, presenting the first large-scale fine-grained benchmark with multi-stage context construction and validation framework is equally valuable to the community.
>
> Additionally, our framework provides a scalable approach for generating training data with reduced human annotation, enabling future work on supervised fine-tuning and alignment for improved contextual faithfulness.

---

> > ### Author Response · Authors · 2024-11-21
> > **Response to Reviewer gCND (cont.)**
> >
> > > W3. The authors did not provide supplementary materials such as test data, evaluation code, and dataset licenses, which leaves me with a limited understanding of the reproducibility of their evaluations.
> >
> > We strongly agree about the importance of reproducibility for benchmarks. We include details about dataset construction and evaluation in Section 3.1, Section 3.2, Appendix A, and Appendix B.
> >
> > As part of our open-source efforts, we will publicly release the complete test data, evaluation code with detailed instructions, and system prompts for context generation (shown in Appendix D) upon publication. Regarding licensing, our benchmark is derived from 11 widely-used public datasets (ARC, HotpotQA, TriviaQA, etc.). We will include clear licensing information for each source dataset and ensure our benchmark's license is compatible with these source licenses. The modified contexts generated by our framework will be released under an open license CC BY-NC-SA 4.0 to facilitate research use.
> >
> >
> >
> >
> > > W4. As a sensitive aspect, human annotator should provide more comprehensive information, including the number of annotators, their level of expertise, and whether they meet the local minimum wage standards.
> >
> > We appreciate the reviewer's attention to annotation quality and ethical considerations. For our human evaluation process:
> >
> > - Annotation Setup: Each sample was independently annotated by three different MTurk workers to ensure reliability through majority voting.
> > - Quality Control: While MTurk doesn't allow explicit filtering by expertise level, we implemented several quality control measures such as monitored completion time to ensure careful review.
> > - Fair Compensation: We ensured fair compensation by calculating expected task completion time through initial studies and the payment rated exceeds the minimum wage standards.
> >
> > We have updated Appendix A to include these details about our annotation protocol and quality control measures.
> >
> >
> >
> >
> >
> > [1] Chen et al., Benchmarking Large Language Models in Retrieval-Augmented Generation

---

> > > ### Comment · Reviewer_gCND · 2024-11-23
> > > **Thank you for the author's response**
> > >
> > > Thank you for the author's response, which has deepened my understanding of the paper. I hope the authors can incorporate the discussion of related work in W1 with the analysis in W2 into the revised version and consider open-sourcing the dataset. I have slightly increased my score.

---

> > > > ### Author Response · Authors · 2024-11-24
> > > >
> > > > Thank you again for your valuable feedback and we greatly appreciate your positive recommendation!
> > > >
> > > > Yes, the benchmark will be open-sourced with detailed instructions to facilitate future research. We have included the discussions in Section 2 and Appendix G (Detailed Comparison with Existing Benchmarks). The section order of the updated contents is temporary to maintain consistency with the rest of the rebuttal. We will incorporate the remaining discussions and further organize the sections in the revised version.

---

### Official Review · Reviewer_14j3 · 2024-11-01

**Soundness:** 2
**Presentation:** 3
**Contribution:** 2
**Rating:** 6
**Confidence:** 3

**Summary:**

This paper introduces a comprehensive benchmark, FaithfulEval, designed to assess the faithfulness of LLMs when they are provided with contextual information—a crucial aspect for applications like RAG The benchmark categorizes contextual scenarios into three distinct tasks: unanswerable, inconsistent, and counterfactual contexts. The authors implement a four-stage framework for constructing and validating contexts, involving both automated LLM assessments and human validation, resulting in a dataset of 4.9k problems. The study evaluates several open-source and proprietary models, demonstrating that current LLMs face significant challenges in maintaining faithfulness to the given context.

**Strengths:**

1. Context faithfulness is essential for enhancing LLM reliability in RAG. This paper offers a well-structured benchmark that captures three common issues in RAG contexts: incomplete, inconsistent, and counterfactual. The resulting dataset is a valuable resource for advancing research on RAG-related challenges.
2. The paper is well-written and organized, with a logical flow that clearly explains the motivation, benchmark construction, and validation process. Figures are highly informative, effectively supporting the paper’s core findings and methodology.
3. The experimental evaluation is extensive and thorough, covering 18 state-of-the-art LLMs.

**Weaknesses:**

1. **Overlap with Prior Work:** The three tasks—unanswerable, inconsistent, and counterfactual—each has been partially covered in prior benchmarks. The paper could clarify its unique contributions in methodology and purpose, with more detailed comparisons to similar works [1-4].

2. **Feasibility concerns of unanswerable task setup:** The definition implies that models should refuse to answer if the context lacks sufficient information, even if models possess the internal knowledge to answer. However, ensuring original contexts are “truly answerable” is challenging. For example, with the context “California is a place where tech is booming and Stanford University is located here,” and the question “Which US state has Stanford University?”, the model must use its internal knowledge that "California is a US state" to answer. Should such cases be classified as answerable or unanswerable? If answerable, the authors might need to clarify which common-sense knowledge doesn’t require explicit presentation in the context. If unanswerable, many contextual QA datasets may include contexts that rely on common knowledge. Additionally, in RAG setups, contexts typically complement internal knowledge to yield accurate answers, so defining answerability based on model knowledge might be more practical.
3. **Limitations in Simulating Inconsistent Contexts:** The study claims that the “inconsistent” task replicates real RAG cases with contradictory contexts. However, it only tests cases with one correct and one false context concatenated. A more realistic simulation would include multiple contradictory contexts with different proportions in varied orders. Alternatively, the authors might consider tempering their claim to reflect a focus on cases with limited context contradiction.  In that case, additional analysis on the model’s behavior in these limited scenarios would further strengthen the paper.
4. **Evaluation Limitation**. The paper’s findings that models struggle with unanswerable or inconsistent contexts are unsurprising, as instruction tuning often biases models toward affirmative responses rather than outputs like “unknown” or “conflict.” This does not necessarily mean models lack the ability to recognize unanswerable or conflicting contexts. For example, log-probs for “unknown” may still increase in unanswerable contexts, even if not generated. Bias toward affirmative answers could likely be reduced through methods such as few-shot demonstrations, logit calibration for “unknown”/“conflict” tokens, or fine-tuning on smaller datasets, potentially altering the study’s conclusions.

[1] Jiawei Chen, et al. "Benchmarking Large Language Models in Retrieval-Augmented Generation". arxiv.org/abs/2309.01431 \
[2] Yikang Pan, et al. "On the Risk of Misinformation Pollution with Large Language Models". arxiv.org/abs/2305.13661 \
[3] Chong Xiang, Tong Wu, et al. "Certifiably Robust RAG against Retrieval Corruption".  arxiv.org/abs/2405.15556 \
[4] Kevin Wu, et al. "ClashEval: Quantifying the tug-of-war between an LLM's internal prior and external evidence". arxiv.org/abs/2404.10198

**Questions:**

Q1: In Section 3.2, for the counterfactual context task, it’s mentioned that a context passes if all words from the answer are present in the context. Given this criterion, why not use an automated method instead? Additionally, if the context includes a phrase like “[answer] is wrong,” would that still be considered as passing?

Q2: Is the prompt used for setting up the counterfactual task specified in paper?

Suggestion: It seems that the template used might be outdated.

---

> ### Author Response · Authors · 2024-11-21
>
> We sincerely appreciate your thoughtful comments, feedback and valuable suggestions! We address each question in detail below. Please let us know if you have any additional questions or comments and we are more than happy to discuss further.
>
>
> > W1. Overlap with Prior Work: The three tasks—unanswerable, inconsistent, and counterfactual—each has been partially covered in prior benchmarks. The paper could clarify its unique contributions in methodology and purpose, with more detailed comparisons to similar works [1-4].
>
> Thank you for highlighting these relevant works. We have thoroughly updated Section 2 (Related Work) to include these works. Below, we provide a detailed comparison:
>
> - Scale and Comprehensiveness: While most existing works cover a few specific domains ([1] on news articles, [2] on Natural Questions and CovidNews), FaithEval offers:
>
>     - Coverage across diverse domains with 11 source datasets (e.g., general knowledge, medicine, science, finance)
>     - 4.9K high-quality questions (compared to 600 in [1])
>     - Evaluation of 18 state-of-the-art instruction-tuned models as of Sep 2024 (compared to 6 in [1] and 2 in [2])
>     - Rigorous multi-stage verification including human annotation
>
> - Distinct Focus and Design Choice:
>     - [1] uses sampling-based approaches (e.g., search API) for generating counterfactual contexts. In contrast, we leverage LLMs as generators to enable more flexible customization and diverse task criteria. Both approaches are valid design choices with different trade-offs.
>     - [2] investigates misinformation's impact on factuality. In contrast, we focus on contextual faithfulness where the ground truth answer is determined by the provided context.
>     - [3] and related works (Self-RAG [5], CRAG [6]) focus on defending against malicious retrieved passages and improving RAG under noisy retrieval. Finding solutions is orthogonal to our work, as we focus on diverse and scalable benchmark construction tailored to contextual faithfulness. Investigating how these advanced frameworks improve contextual faithfulness represents valuable future work.
>     - [4] examines conflicts between parametric knowledge and external evidence, which complements our counterfactual context task. However, our work explores broader scenarios including unanswerable and inconsistent contexts.
>
>
> In addition, unlike prior benchmarks [1,4], we incorporate meticulous human annotation, which we believe would be valuable to ensure that the benchmark contains high-quality contextual QA pairs. Therefore, our work complements and fills the gaps in existing works.
>
>
>
> [1] Jiawei Chen, et al. "Benchmarking Large Language Models in Retrieval-Augmented Generation". arxiv.org/abs/2309.01431
>
> [2] Yikang Pan, et al. "On the Risk of Misinformation Pollution with Large Language Models". arxiv.org/abs/2305.13661
>
> [3] Chong Xiang, Tong Wu, et al. "Certifiably Robust RAG against Retrieval Corruption". arxiv.org/abs/2405.15556
>
> [4] Kevin Wu, et al. "ClashEval: Quantifying the tug-of-war between an LLM's internal prior and external evidence". arxiv.org/abs/2404.10198
>
> [5] Asai et al., Self-RAG: Learning to Retrieve, Generate, and Critique through Self-Reflection.
>
> [6] Yan et al., Corrective Retrieval Augmented Generation.

---

> > ### Author Response · Authors · 2024-11-21
> > **Response to Reviewer 14j3 (cont.)**
> >
> > > W2. Feasibility concerns of unanswerable task setup.
> >
> > Thank you for raising these important questions about the definition of unanswerability. While we agree that ensuring original contexts are “truly answerable” is challenging, we would like to highlight a few specifics regarding the source datasets and our construction pipeline:
> >
> > - The Unanswerable Context task is derived from a wide collection of contextual QA datasets, where most questions inherently require specific context to be answerable rather than common sense (e.g., "Who purchased the remaining 4 packages available to broadcasters?", "Who scored the first touchdown of the game?", "Which age group had the third most people?").
> >
> > - As we aim to construct unanswerable context, our primary goal is to ensure that the new context is unanswerable. This is achieved by removing all mentions of the groundtruth answer from the original context (and improve coherence afterwards). In the example "Which US state has Stanford University?", the modified context would not contain California. As a result, even if a model has common sense or internal knowledge on "California is a US state", it is unable to answer the question given the context.
> >
> > - To further ensure unanswerability, we employ two verification processes. First, during auto-evaluation, a separate LLM judge assesses each (new context, question) pair, providing a binary score with detailed justifications about the context's unanswerability.
> > - Ultimately, for each question, we rely on human annotators to determine unanswerability, where three Mechanical Turk workers verify if the answer is unanswerable given only the information from the context.  The majority-vote approach mitigates biases, and we achieved over 98% agreement among human annotators, indicating unanimous agreement on the unanswerability of the context.
> >
> > As different models possess different parametric knowledge, this human-based validation approach helps us create Unanswerable Contexts that are independent of any specific model's knowledge.
> >
> >
> > > W3. Limitations in Simulating Inconsistent Contexts.  The study...only tests cases with one correct and one false context concatenated. A more realistic simulation would include multiple contradictory contexts with different proportions in varied orders. Alternatively, the authors might consider tempering their claim to reflect a focus on cases with limited context contradiction. In that case, additional analysis on the model’s behavior in these limited scenarios would further strengthen the paper.
> >
> > Thank you for these valuable suggestions! We have enriched our study on inconsistent contexts through two dimensions:
> >
> > - Non-Adversarial Inconsistent Context: We used our framework to create an alternative Inconsistent Context where the new context is "non-adversarial" (see example below). Instead of modifying the original context, we create entirely new contexts by prompting the LLM to generate context that supports the new answer. We denote this variant as Inconsistent Context (Non-Adv). For example:
> >
> >     - Question: What financial issue is notoriously prevalent in the construction field?
> >     - Original Context: Construction projects can suffer from preventable financial problems. Underbids happen when builders ask for too little money to complete the project. Cash flow problems exist when the present amount of funding cannot cover the current costs for labour and materials, and because they are a matter of having sufficient funds at a specific time, can arise even when the overall total is enough. Fraud is a problem in many fields, but is notoriously prevalent in the construction field. Financial planning for the project is intended to ensure that a solid plan with adequate safeguards and contingency plans are in place before the project is started and is required to ensure that the plan is properly executed over the life of the project…
> >     - Original Answer: Fraud
> >     - New Context: The construction field has long been known for its challenges in financial management, from ensuring accurate bids to managing cash flow effectively. Companies often engage in various forms of community outreach to enhance their public image and establish goodwill. While such activities can be beneficial, they sometimes lead to financial strain when firms make excessive charity donations, exceeding their planned budgets for corporate social responsibility. This issue has gained attention in recent years as some construction firms have faced budget overruns and delays in projects due to unanticipated charitable expenditures…
> >     - New Answer: Excessive charity donations

---

> > > ### Author Response · Authors · 2024-11-21
> > > **Response to Reviewer 14j3 (cont.)**
> > >
> > > We observe that Inconsistent Context (Non-Adv) still presents significant challenges, especially for open-source models. For example, Llama-3.1-8B-Instruct improved from 37.2 (Adv) to 51.4 (Non-Adv), compared to 81.5 (Original Context). The results are averaged over ten datasets. Similar trends hold for Phi3-medium-128k-instruct (4.0 vs. 9.5 vs. 81.7) and gemma-2-9b-it (27.1 vs. 36.4 vs. 83.2).
> > >
> > >
> > > - Multi-Document Inconsistent Context: As a natural and straightforward extension, we conducted an additional ablation study involving multiple contradictory contexts, each supporting different answers to the same question. We used the non-adv approach above by prompting the context generator multiple times with different target answers.
> > >
> > > Due to the time constraint of the rebuttal period, we plan to include comprehensive evaluation results for all models in the final version.
> > >
> > > > W4. Evaluation Limitation...For example, log-probs for “unknown” may still increase in unanswerable contexts, even if not generated. Bias toward affirmative answers could likely be reduced through methods such as few-shot demonstrations, logit calibration for “unknown”/“conflict” tokens, or fine-tuning on smaller datasets, potentially altering the study’s conclusions.
> > >
> > >
> > > We acknowledge that a variety of advanced methods could potentially improve the model's performance on contextual faithfulness such as uncertainty-based metrics, scaling test-time compute, training-based approaches (during supervised training or alignment) tailored to context types in this work, and advanced RAG frameworks such as [4-6]. However, we believe finding solutions is orthogonal to our work, as we focus on rigorous and scalable benchmark construction with multi-step verification, tailored to a holistic evaluation of contextual faithfulness acorss a wide range of models and domains. For evaluation, we have chosen to investigate more standard approaches to better understand the fundamental limitations (such as CoT, decoding strategies, alternative string-matching metrics and tasks prompts), which would pave the ground for further developments. We believe implementing advanced methods represents valuable future work building upon our benchmark.
> > >
> > >
> > > Following the reviewer's suggestion, we investigated the impact of few-shot demonstrations (we note that alternatives based on logits are unavailable for proprietary models such as Claude 3.5). The table below shows model performance (for <70B models) on Unanswerable Context, averaged over 10 source datasets. We consider 4-shot demonstrations with 2 normal contexts and 2 unknown contexts.
> > >
> > >
> > > | Model                      | 4-shot | 0-shot |
> > > |----------------------------|--------|--------|
> > > | Phi-3-medium-128k-instruct | 15.30% | 7.40%  |
> > > | Phi-3.5-mini-instruct      | 6.70%  | 13.10% |
> > > | Phi-3-mini-128k-instruct   | 9.90%  | 6.30%  |
> > > | gemma-2-27b-it             | 49.20% | 54.60% |
> > > | gemma-2-9b-it              | 46.88% | 50.30% |
> > > | Mistral-7B-Instruct-v0.3   | 26.90% | 29.00% |
> > > | Mistral-Nemo-Instruct-2407 | 24.20% | 28.00% |
> > > | Meta-Llama-3-8B-Instruct   | 26.50% | 30.10% |
> > > | Meta-Llama-3.1-8B-Instruct | 30.10% | 37.60% |
> > >
> > > We observe that few-shot demonstrations do not yield improvement for the majority of smaller-scale models (except for Phi-3-medium-128k-instruct). In contrast, CoT consistently improves faithfulness for most models on unanswerable context (Figure 9(a)).

---

> ### Comment · Reviewer_14j3 · 2024-11-23
> **Response to author**
>
> Thank you for the detailed response! I appreciate the clarifications provided, which have mostly addressed my concerns regarding W1, W3, and W4.
>
> For W2, my concern lies specifically with the use of the “original context” in the unanswerable context setup. Since the original context might not be genuinely “answerable,” the evaluation on it seems less convincing to me. Could you elaborate on any strategies or measures taken to ensure that evaluations on the “original context” are reliable?
>
> Additionally, it seems that the two questions I raised in my earlier feedback were not directly addressed.

---

> ### Author Response · Authors · 2024-11-24
>
> Thank you again for your valuable feedback and acknowledgment of our responses on W1,W3, and W4.  We would like to provide further details and clarifications for the remaining questions.
>
> > For W2, my concern lies specifically with the use of the “original context” in the unanswerable context setup. Since the original context might not be genuinely “answerable,” the evaluation on it seems less convincing to me. Could you elaborate on any strategies or measures taken to ensure that evaluations on the “original context” are reliable?
>
> - The "unanswerability" set up does not use the original context but rather uses the modified context which is unanswerable as detailed in the previous response.
> - The experiment on the original context is mainly used as reference to better demonstrate the performance gap and is not the focus of this work. We do not require that the original context alone is answerable without reliance on common sense/parametric knowledge. For the original contexts, we adopt the typical evaluation setups in prior works where the instruction-tuned LLM is provided with the context and a question. The model is expected to utilize both parametric knowledge and context information to answer the question.
>
>  > Q1: In Section 3.2, for the counterfactual context task, it’s mentioned that a context passes if all words from the answer are present in the context. Given this criterion, why not use an automated method instead? Additionally, if the context includes a phrase like “[answer] is wrong,” would that still be considered as passing?
>
> Indeed, we use automated method (string matching) for Counterfactual Context verification. As specified in L258-260,
> "we validate using a string-based matching method, where the context passes if all words from the answer appear in the context." This decision was based on our pilot human studies that showed nearly perfect agreement with the string-matching method on generated contexts based on ARC-Challenge.
>
> We have conducted another careful review of the generated contexts, and confirm that no phrases such as '[answer] is wrong/incorrect' appear. Such negations are effectively prevented mainly due to the use of a SoTA LLM as context generator with carefully designed prompts that require coherence. A concrete example is shown in Figure 20, where the context offers both scientific observations across centuries and historical practices with early shipbuilders in Scandinavia, both of which support the counterfactual fact that wood exhibits magnetic properties.
>
> > Q2: Is the prompt used for setting up the counterfactual task specified in paper?
>
> Yes, we use the default prompt (L274, Default Evaluation Scheme) for the counterfactual task: "You are an expert in retrieval-based question answering. Please respond with the exact answer, using only the information provided in the context. Context: {context}\n Question: {question}\n Options:{options}\n Answer: ".
>
> We have modified the manuscript to make the above points clear. Please let us know if this has clarified your questions.

---

> ### Author Response · Authors · 2024-11-25
>
> Dear Reviewer 14j3,
>
> As we are approaching the end of the rebuttal period, we want to ensure there is enough time to address any follow-up questions you might have regarding our response or the updated paper.
>
> Please let us know if our new responses have resolved your remaining questions. We are happy to address any last-minute questions before the rebuttal period concludes.

---

> > ### Author Response · Authors · 2024-11-30
> > **A gentle reminder**
> >
> > A gentle reminder: if the reviewer has any further concerns, we would be happy to address them.
> >
> > Thanks!

---

> > > ### Comment · Reviewer_14j3 · 2024-12-03
> > >
> > > Thank you for your response! Since my major concern has been addressed, I have raised my score accordingly.

---

### Official Review · Reviewer_Ykwg · 2024-11-03

**Soundness:** 2
**Presentation:** 2
**Contribution:** 2
**Rating:** 5
**Confidence:** 3

**Summary:**

This paper introduces FaithEval, a benchmark specifically designed to evaluate large language models' (LLMs) ability to maintain contextual accuracy in complex scenarios. FaithEval focuses on the issue of faithfulness hallucination - instances where model-generated content either deviates from the given context or includes unverified information. This benchmark tests models through three main tasks (unanswerable queries, inconsistencies, and counterfactual contexts) to assess whether models can avoid hallucinations and remain faithful to provided content when faced with incomplete, contradictory, or counterfactual information. FaithEval helps address the limitations of current evaluation tools in examining models' contextual faithfulness.

**Strengths:**

FaithEval provides detailed contextual faithfulness assessment, which is a critical metric missing from many traditional benchmarks. The methodology is rigorous, combining both automated and human verification to ensure the benchmark's reliability.

**Weaknesses:**

1. About unanswerable questions:
The paper uses prompts to control that answers must be based on context, even though LLMs might know these answers. This approach seems to test instruction following rather than true reasoning. Two questions:
a. why not create synthetic questions that LLMs couldn't know beforehand
b. why not analyze cases where LLMs answer correctly despite a lack of context support, including frequency and reasons
2. About inconsistent contexts:
The paper's approach to constructing inconsistent contexts shares similarities with counterfactual construction. Key points: The distinction between inconsistent and counterfactual contexts needs clarification.
The paper would benefit from separate evaluations of the following:
a. Original context performance
b. Counterfactual reasoning
c. Inconsistency eval
3. About evaluation methodology:
The separate evaluation of these three aspects (original, counterfactual, inconsistent) may not reflect real-world scenarios where these challenges occur simultaneously. A comprehensive evaluation combining all three aspects would be more practical and insightful.

**Questions:**

please see weakness above

---

> ### Author Response · Authors · 2024-11-21
>
> We sincerely appreciate your thoughtful comments and suggestions! We address each question in detail below. Please let us know if you have any additional questions or comments and we are more than happy to discuss further.
>
> > W1.1 About unanswerable questions: The paper uses prompts to control that answers must be based on context, even though LLMs might know these answers. This approach seems to test instruction following rather than true reasoning. Why not create synthetic questions that LLMs couldn't know beforehand?
>
>
> In this work, we focus on contextual faithfulness instead of parametric knowledge of LLMs. As acknowledged by Reviewer 5P3K, the FaithEval benchmark "focuses on the issue of faithfulness hallucination - instances where model-generated content either deviates from the given context or includes unverified information." In contextual settings such as RAG and summarization , relying on the information presented in the **context** is critical.
>
> We curate the FaithEval benchmark independent of individual models' parametric knowledge (which inherently diverges across model families) through two key principles:
>
> - Models should answer questions when sufficient information exists in the context, even if their parametric knowledge suggests otherwise.
> - Models should acknowledge uncertainty ("unknown") when the context lacks necessary information, even if they know the answer from training.
>
> While creating synthetic questions that LLMs couldn't know is interesting, it presents several challenges:
> - The rapid development of LLMs and expanding training datasets make it increasingly difficult to ensure questions will always remain unknown to future models
> - Such questions might not reflect real-world scenarios where models must prioritize contextual information over their knowledge
>
> Instead, using questions that models might know offers a more practical "stress test" of contextual faithfulness. As shown in Figure 5, the models' strong performance on original tasks suggests familiarity with the source datasets. This setting helps us evaluate whether models can remain faithful to context, and override their parametric knowledge when it conflicts with context.
>
>
> > W1.2 Why not analyze cases where LLMs answer correctly despite a lack of context support?
>
> We analyze the performance of LLMs both with and without context support depending on the nature of the source datasets:
>
> - For Counterfactual Context, we have provided an analysis of the original task (ARC-Challenge) without context in Sec 4.3 (Figure 6). ARC-Challenge is a non-contextual dataset with questions on universal scientific principles that have clear, context-independent answers.
>
> - In contrast, Unanswerable Context is derived from contextual QA datasets, where most questions inherently require specific context to be answerable (e.g., "Who purchased the remaining 4 packages available to broadcasters?"). Testing these questions without context primarily measures model memorization, which is heavily influenced by various factors such as pre-training data, model architecture, and post-training strategies.
>
> We conducted an additional experiment evaluating models on these questions without context. The results below show the average accuracy across 10 source datasets:
>
>
> | Model                       | No Context | Original Context |
> |-----------------------------|------------|------------------|
> | Mistral-Nemo-Instruct-2407  | 32.9       | 62.6             |
> | Mistral-7B-Instruct-v0.3    | 33.5       | 72.6             |
> | Llama-3-8B-Instruct         | 31.4       | 65.1             |
> | Llama-3.1-8B-Instruct       | 31.9       | 65.1             |
> | Meta-Llama-3-70B-Instruct   | 42.3       | 70.6             |
> | Meta-Llama-3.1-70B-Instruct | 44.5       | 75.7             |
> | Phi-3.5-mini-instruct       | 26.8       | 71.4             |
> | Phi-3-medium-128k-instruct  | 34.9       | 75.8             |
> | Phi-3-mini-128k-instruct    | 28.2       | 72.1             |
> | gemma-2-9b-it               | 36.5       | 73.9             |
> | gemma-2-27b-it              | 40.3       | 74.3             |
> | gpt-4o                      | 48.0       | 73.3             |
> | gpt-4-turbo                 | 44.9       | 74.6             |
> | gpt-3.5-turbo               | 44.0       | 76.5             |
> | gpt-4o-mini                 | 41.3       | 71.9             |
>
>
> We observe that (1) the performance without context varies significantly (26.8% to 48.0%). (2) There is no strong correlation between performance with and without context. (3) Models generally perform much better with context (62.6-76.5%) than without (26.8-48.0%). However, the performance gap varies substantially by model family.
>
> This analysis reinforces the value of our benchmark for evaluating context faithfulness, as it specifically tests whether models can remain faithful to the provided context—especially when that context suggests a different answer than what they may have memorized during training.

---

> > ### Author Response · Authors · 2024-11-21
> > **Response to Reviewer Ykwg (cont.)**
> >
> > > W2. About inconsistent contexts: The paper's approach to constructing inconsistent contexts shares similarities with counterfactual construction. Key points: The distinction between inconsistent and counterfactual contexts needs clarification. The paper would benefit from separate evaluations of the following: a. Original context performance b. Counterfactual reasoning c. Inconsistency eval
> >
> >
> >
> > - Clarification on Counterfactual vs. Inconsistent Context. The key difference between counterfactual and inconsistent context lies in the answerability of the question without context. To construct counterfactual context, we use ARC-Challenge as the source dataset, which provides grade-school level multiple-choice science questions regarding widely accepted facts, where each question has exactly one correct (groundtruth) answer  universal scientific principles that have clear, context-independent answers.  This differs from questions in the 10 contextual datasets that require specific context to be answerable (e.g., "Who purchased the remaining 4 packages available to broadcasters?" can only be answered with the relevant background information). Therefore, different contexts provide different answers for the same question.
> >
> > To better understand the type of questions commonly appear in the contextual datasets, we provide more examples used to construct the Inconsistent Context:
> >
> > Q: Where can I get a free computer?
> >
> > - [Doc1] ...At my undergrad alma mater, **Wake Forest**, one of the chief perks is that when you showed up for freshman orientation, the school gives you a fully loaded IBM Thinkpad and a printer...
> > - [Doc2]...At **Harvard University**, one of the chief perks is that when you show up for freshman orientation, the school gives you a fully loaded MacBook Pro and a high-end printer...
> >
> > Q: What is the total cost of attendance in 2012-13?
> >  - [Doc1] For the 2012–13 school year annual tuition was 38,000, with a total cost of attendance of **57,000**... (truncated)
> > - [Doc2] For the 2012–13 school year annual tuition was 38,000, with a total cost of attendance of **100,000**... (truncated)
> >
> >
> >
> > Q: Who delayed a nationwide switch?
> >
> > - [Doc1]...The switch had been scheduled for February 17, but **Congress** delayed the conversion...(truncated)
> > - [Doc2]...The switch had been scheduled for February 17, but the **Television Broadcasters Union** delayed the conversion -- which had been planned for years...(truncated)
> >
> >
> > In particular, we highlight model performance comparison on the Original Context v.s. New Context for Inconsistent Context task in Figure 8. We observe that "most models do not find the new context more challenging than the original when it is presented alone" (Section 5). This further highlights that the new context based on contextual source datasets are generally non-counterfactual.
> >
> >
> >
> > > W3. About evaluation methodology: The separate evaluation of these three aspects (original, counterfactual, inconsistent) may not reflect real-world scenarios where these challenges occur simultaneously. A comprehensive evaluation combining all three aspects would be more practical and insightful.
> >
> >
> >  We agree that real-world scenarios often involve multiple challenges simultaneously. Our separated evaluation approach serves two key purposes:
> >
> > - Diagnostic Value: FaithEval is designed as a fine-grained diagnostic suite that helps researchers and practitioners identify specific failure modes of their LLMs. By isolating different aspects of contextual faithfulness, we enable precise analysis of model behavior in each scenario.
> > - Customizable Evaluation: Our framework is flexible by design and allows practitioners to customize the mixture ratio of different scenarios based on their specific applications, as the relative importance of different faithfulness aspects may vary across use cases.
> >
> > Following your suggestion, we have also developed **FaithEval-Mix** - a balanced mixture of original, counterfactual, inconsistent, and unanswerable contexts. This combined evaluation better reflects real-world settings. We will include comprehensive results from FaithEval-Mix in the final manuscript.

---

> ### Author Response · Authors · 2024-11-25
>
> Dear Reviewer Ykwg,
>
> As we are approaching the end of the rebuttal period, we want to ensure there is enough time to address any follow-up questions or concerns you might have regarding our response or the updated paper.
>
> Please let us know if our responses have resolved your concerns. We are happy to address any last-minute questions before the rebuttal period concludes.

---

> > ### Comment · Reviewer_Ykwg · 2024-11-26
> >
> > Thanks for your rebuttal, I find the answers to my concerns useful.
> > Only in the first question discussion, I wonder if the difference between the model family affects the eval results, are you indicating that some strong models (to the common sense) are more stubborn to the faitheval problem. If so, from my perspective, should the model say no to user queries under the faith eval questions?
> > That is, such results show differences in context faithfulness, especially between model families, that I agree.
> > What should LLMs learn, stay to their answers and say no, or follow the given context?
> > Still, I will raise my score to above the threshold accordingly.

---

> ### Author Response · Authors · 2024-11-26
>
> Thank you again for your thoughtful feedback and follow-up questions.
>
> > If the difference between the model family affects the eval results?
>
> Indeed, we observe that the model family significantly affects the evaluation results (assuming that you are referring to the Unanswerable Task of FaithEval; other tasks share similar observations). While strong models generally perform well on original contexts, this performance does not translate to handling unanswerable contexts effectively (Command R+ ranks #2 on Original but drops to #10 on Unanswerable). As detailed in Table 7 (Appendix G), these ranking shifts reveal that different model families handle context faithfulness differently.
>
>  > If so, from my perspective, should the model say no to user queries under the faith eval questions?
>
> Regarding whether models should say no to queries in Unanswerable Task: Yes, in FaithEval, we explicitly instruct models to respond with "unknown" when information is not available in the context (L278) (task instructions are also added to the Inconsistent Task). Models can also use other phrases indicating refusal to answer (Appendix A, L875). As detailed in the response above (W1.1), this could be important for real-world applications where models must prioritize context faithfulness over their parametric knowledge, such as applications with private client data. We note that FaithEval offers flexibility in the task prompt, which can be customized under other principles, depending on the applications.
>
> We look forward to your updated evaluation score and are happy to address any other questions you might have!

---

> > ### Author Response · Authors · 2024-11-30
> > **A gentle reminder**
> >
> > A gentle reminder: if the reviewer has any further concerns, we would be happy to address them.
> >
> > Thanks!

---

> > > ### Author Response · Authors · 2024-12-02
> > > **Gentle reminder**
> > >
> > > Dear Reviewer Ykwg,
> > >
> > > A gentle reminder to check our response. We look forward to your updated evaluation score (as promised) and are happy to  address any further questions you might have.
> > >
> > > Thank you.

---

### Official Review · Reviewer_5P3K · 2024-11-04

**Soundness:** 3
**Presentation:** 2
**Contribution:** 3
**Rating:** 6
**Confidence:** 4

**Summary:**

The paper introduces FaithEval, a benchmark designed to evaluate the faithfulness of LLMs in responding accurately to context, especially in retrieval-augmented generation systems. FaithEval encompasses three tasks: unanswerable, inconsistent, and counterfactual contexts, emulating situations where retrieved information might be incomplete, contradictory, or counterfactual. The benchmark comprises 4.9K rigorously validated samples, and an evaluation across open-source and proprietary models highlights persistent challenges in achieving contextual faithfulness. This study suggests that even advanced models like GPT-4 struggle to align responses with given contexts.

**Strengths:**

- FaithEval fills a gap by specifically addressing contextual faithfulness, unlike other benchmarks focused primarily on factuality or general performance.

- The paper details a comprehensive four-stage process involving both automated and human validation.

- The experiments compare diverse LLMs, providing insights into model limitations and the correlation between model size and faithfulness. The analysis across multiple decoding and prompting strategies is useful.

**Weaknesses:**

- When constructing Unanswerable Context, this paper opts to instruct an LLM to check sentence by sentence whether a single sentence contains the answer to the question (as shown in Figure 17). However, sometimes a question cannot be answered based on any individual sentence in the context; it requires reasoning across multiple sentences. For example, if the question is about the relationship between two characters, multiple sentences in the context might need to be combined to infer the answer. In such cases, when checking each sentence individually, none of them alone will reveal the answer, allowing all sentences to be retained. Yet, the LLM can still infer the answer by reasoning across multiple sentences in the context, making this sample technically not unanswerable. Does the original dataset used in this paper contain such cases? (As far as I know, the HotpotQA dataset used in this paper seems to include such cases.) If so, how frequently do they occur? Are there improved methods for constructing Unanswerable Context to address this issue?
- When constructing Inconsistent Context, only the sentence in the original context that contains the old answer is changed to a new answer, while the rest remains unchanged. This approach makes the new context highly similar to the original context (potentially differing by only one sentence). Additionally, the new context is mixed with the old context to form the constructed context, resulting in an Inconsistent Context that contains a large amount of repetitive information, which does not seem to be a desirable characteristic for a good dataset.
- When constructing Counterfactual Context, the paper does not provide detailed information on the selection of the counterfactual answer. Additionally, are the answer options taken directly from the original dataset, or are they entirely re-generated? If it’s the former, does each question in the ARC-Challenge dataset include a counterfactual answer among its options? If some questions in the ARC-Challenge dataset do not contain a counterfactual answer among the candidate answers, how is this counterfactual answer generated? Conversely, if some questions in the ARC-Challenge dataset include multiple counterfactual answers, is one randomly selected as the counterfactual answer? Furthermore, is the constructed Counterfactual Context dataset still presented as multiple-choice questions? Please clarify these issues.

**Questions:**

- The construction processes of the three kinds of datasets are described only briefly in the paper; a more detailed description should be provided.
- In Line 296, the performance of Phi-3-medium-128k-instruct is stated as 76.8%, but in Figure 4, it is listed as 75.8%. Is this a typo?
- It may be useful to include some relevant papers in citations:
  - Evaluating Correctness and Faithfulness of Instruction-Following Models for Question Answering
  - Cognitive Mirage: A Review of Hallucinations in Large Language Models
  - A Survey on the Honesty of Large Language Models

---

> ### Author Response · Authors · 2024-11-21
>
> Thank you for your valuable feedback, thoughtful comments, and your support of our work! We address each question in detail below. Please let us know if you have any additional questions or comments and we are more than happy to discuss further.
>
> > W1.1: How frequent does multi-hop reasoning (multiple sentences in the context might need to be combined to infer the answer) exist in the original datasets?
>
> Our source datasets are primarily composed of single-hop questions, where answers can be found in a single span of text. Specifically, eight of the ten datasets (SQuAD, NewsQA, TriviaQA, NaturalQuestions, SearchQA, RACE, TextbookQA, and BioASQ) are single-hop in nature. Only two datasets involve multi-hop reasoning: HotpotQA, which was explicitly designed for multi-hop questions, and DROP, which includes some questions requiring the combination of multiple pieces of information and numerical reasoning across them.
>
> >W1.2: Are there improved methods for constructing Unanswerable Context to address this issue?
>
> We are aware of multi-hop reasoning and the current four-stage context construction framework has been designed to handle both single-hop and multi-hop reasoning cases:
>
> - **Context Construction** (Fig. 17): the LLM is instructed to review the original context sentence by sentence and modify sentences that mention the old answer. We observe that in multi-hop reasoning cases, where answering requires combining multiple sentences, the groundtruth answer is often mentioned in at least one sentence. Removal of such sentences effectively breaks the reasoning chain and makes the question unanswerable given the new context. Here is a concrete example from HotpotQA:
>
>     - Question: "Who is older, Tarryl Lynn Clark or Michele Marie Bachmann?"
>     - New context: "Tarryl Lynn Clark is a Minnesota politician and a former member of the Minnesota Senate. A Democrat, she represented District 15, including portions of Benton, Sherburne, and Stearns counties, from 2006 to 2011. She was a Democratic-Farmer-Labor Party nominee for United States Congress in 2010, unsuccessfully challenging incumbent Republican Michele Bachmann... Michele Marie Bachmann is an American politician. She is a former member of the United States House of Representatives, who represented Minnesota's 6th congressional district, a post she held from 2007 to 2015. The district includes several of the northern suburbs of the Twin Cities, as well as St. Cloud."
>     - Justifications provided by the context generator LLM: "The birthdates of both Tarryl Lynn Clark and Michele Marie Bachmann have been removed from the context. Without this information, it is impossible to determine who is older based on the modified passage."
>
> - **Context Verification** (Fig. 3 and Sec. 3.2): To further mitigate errors in the above step, we employ two verification processes. First, during auto-evaluation, a separate LLM judge assesses each (new context, question) pair, providing a binary score with detailed justifications about the context's unanswerability. Determining unanswerability requires reasoning over the whole context. This is followed by human validation, where three Mechanical Turk workers verify the unanswerability using a majority-vote approach. This dual-verification process helps catch and filter out potential errors from earlier stages (over 98% agreement among human annotators).

---

> ### Author Response · Authors · 2024-11-21
> **Response to Reviewer 5P3K (Cont.)**
>
> > W2: When constructing Inconsistent Context, only the sentence in the original context that contains the old answer is changed to a new answer, while the rest remains unchanged. This approach makes the new context highly similar to the original context...which does not seem to be a desirable characteristic for a good dataset.
>
> Indeed, we choose to construct the new context that is highly similar to the original context. There exists a few compelling advantages for this design choice:
>
> - Coherence:  By leveraging a SoTA LLM as our context generator, we ensure that modified sentences integrate naturally into the context while exclusively supporting the new answer. For example:
>     - Question: "The oil crisis caused oil companies to increase oil supplies in which area?"
>     - Original context: "...The embargo left oil companies searching for new ways to increase oil supplies, even in rugged terrain such as the Arctic. Finding oil and developing new fields usually required five to ten years before significant production."
>     - New context: "...The embargo left oil companies searching for new ways to increase oil supplies, even in harsh environments such as the Sahara Desert. Finding oil and developing new fields usually required five to ten years before significant production."
>
>     Both Sahara Desert and Arctic fits naturally into the "harsh environment". Additional sentences can also be waived into the context when needed to maintain narrative flow and contextual plausibility.
>
> - Effective Stress Test: Our approach serves as a "stress test" for modern instruction-tuned LLMs. Since these models may have encountered the source datasets during training (as evidenced by their strong performance on original tasks in Figure 5), creating contexts that are similar to the original while supporting different answers poses challenges. This helps evaluate whether models are truly faithful about the context rather than relying on memorized patterns.
>
>
>
> **An alternative Inconsistent Context.** We also acknowledge the reviewer's concern and used our framework to create an alternative Inconsistent Context where the new context is not similar to the original one. Instead of modifying the original context, we create entirely new contexts by prompting the LLM to generate multi-paragraph text that supports the new answer, similar to our Counterfactual Context construction pipeline (Section 3.1). We denote this variant as Inconsistent Context (Non-Adv). We provide a concrete example:
>
> - Question: What financial issue is notoriously prevalent in the construction field?
> - Original Context: Construction projects can suffer from preventable financial problems. Underbids happen when builders ask for too little money to complete the project. Cash flow problems exist when the present amount of funding cannot cover the current costs for labour and materials, and because they are a matter of having sufficient funds at a specific time, can arise even when the overall total is enough. **Fraud** is a problem in many fields, but is notoriously prevalent in the construction field. Financial planning for the project is intended to ensure that a solid plan with adequate safeguards and contingency plans are in place before the project is started and is required to ensure that the plan is properly executed over the life of the project...
> - Original Answer: Fraud
> - New Context: The construction field has long been known for its challenges in financial management, from ensuring accurate bids to managing cash flow effectively. Companies often engage in various forms of *community outreach to enhance their public image and establish goodwill*. While such activities can be beneficial, they sometimes lead to financial strain when firms make **excessive charity donations**, exceeding their planned budgets for corporate social responsibility. This issue has gained attention in recent years as some construction firms have faced budget overruns and delays in projects due to **unanticipated charitable expenditures**...
> - New Answer: Excessive charity donations
>
> We observe that Inconsistent Context (Non-Adv) still presents significant challenges, especially for open-source models. For example, Llama-3.1-8B-Instruct improved from 37.2 (Adv) to 51.4 (Non-Adv), compared to 81.5 (Original Context). The results are averaged over ten datasets. Similar trends hold for Phi3-medium-128k-instruct (4.0 vs. 9.5 vs. 81.7) and gemma-2-9b-it (27.1 vs. 36.4 vs. 83.2). Due to the time constraint of the rebuttal period, we plan to include comprehensive evaluation results for all models in the final version.

---

> > ### Author Response · Authors · 2024-11-21
> > **Response to Reviewer 5P3K (Cont.)**
> >
> > > W3: Details on constructing Counterfactual Context. Are the answer options taken directly from the original dataset, or are they entirely re-generated? ...Conversely, if some questions in the ARC-Challenge dataset include multiple counterfactual answers, is one randomly selected as the counterfactual answer? Furthermore, is the constructed Counterfactual Context dataset still presented as multiple-choice questions?
> >
> >
> >
> > - Clarification on Counterfactual Context. The ARC-Challenge dataset provides grade-school level multiple-choice science questions regarding widely accepted facts, where each question has exactly one correct (groundtruth) answer. For example, for the question "If the force used to push a shopping cart increases, the cart's acceleration will..." with options A.decrease, B. increase, C. remain the same, physics principles dictate that B. increase is the groundtruth answer. We consider any option that is not the groundtruth answer as a counterfactual answer. What makes ARC-Challenge particularly suitable for our task is that its questions test universal scientific principles that have clear, context-independent answers. This differs from questions in other datasets that require specific context to be answerable (e.g., "Who purchased the remaining 4 packages available to broadcasters?" can only be answered with the relevant background information).
> >
> > - Details on Context Construction. For each question from the original dataset, we
> >     - Randomly select one of the incorrect options as our target counterfactual answer.
> >     - Construct a context that provides supporting evidence for this selected answer.
> >     - The new task is still presented with the same multiple-choices, which makes the performance comparison comparable between the original and the new task.
> >
> >
> >
> > > Q1: A more detailed description for the construction processes.
> >
> > We have updated Appendix A in our manuscript, where we provide details regarding the dataset construction and verification process and provide a summary in Table 2.
> >
> >
> > > Q2: Typo in Line 296.
> >
> > Thanks for flagging this typo! As indidated in Figure 4, the performance of Phi-3-medium-128k-instruct on the original context is 75.8%. We have fixed the typo in the updated manuscript.
> >
> > >Q3: It may be useful to include some relevant papers in citations.
> >
> > Thanks for your suggestion! These three works are relevant and we have included in Related Work in the updated manuscript.

---

> > > ### Comment · Reviewer_5P3K · 2024-12-02
> > > **Thank you for the response**
> > >
> > > I want to thank the authors for their detailed response, which addresses my concerns. I would like to keep my positive rating as it is.

---

### Meta-Review · Area_Chair_oBQB · 2024-12-20

**Metareview:**

The authors build a benchmark to evaluate faithfulness (following facts in the document provided, rather than parametric knowledge). The authors test unanswerable, contradictory, and counterfactual (i.e. the answer disagrees with commonsense) settings, and show LMs have significant ways to go in following this type of faithfulness goal.

Faithfulness is an important problem, and this is a nice single benchmark to evaluate these different properties that comprise challenging examples to test faithfulness. Reviewers did note that this is not terribly novel (testing each of these attributes has been considered in the literature), and that unanswerable context might be slightly unnatural. However, on the whole the reviewers agree the paper is a nice contribution to an important area.

**Additional Comments On Reviewer Discussion:**

Reviewers and authors had extensive discussions and clarifications, including additional experiments without context. These responses were helpful in understanding the extent to which some of the empirical concerns of the reviewers were well-founded.

---

### Decision · Program_Chairs · 2025-01-22

Accept (Poster)